# Age-specific differences in the dynamics of protective immunity to influenza

Sylvia Ranjeva[1], Rahul Subramanian[1], Vicky J. Fang[2], Gabriel M. Leung[2], Dennis K.M. Ip[2], Ranawaka A.P.M. Perera [2], J.S. Malik Peiris[2], Benjamin J. Cowling [2] & Sarah Cobey[1]

Influenza A viruses evolve rapidly to escape host immunity, causing reinfection. The form and duration of protection after each influenza virus infection are poorly understood. We quantify the dynamics of protective immunity by fitting individual-level mechanistic models to long-itudinal serology from children and adults. We find that most protection in children but not adults correlates with antibody titers to the hemagglutinin surface protein. Protection against circulating strains wanes to half of peak levels 3.5–7 years after infection in both age groups, and wanes faster against influenza A(H3N2) than A(H1N1)pdm09. Protection against H3N2 lasts longer in adults than in children. Our results suggest that influenza antibody responses shift focus with age from the mutable hemagglutinin head to other epitopes, consistent with the theory of original antigenic sin, and might affect protection. Imprinting, or primary infection with a subtype, has modest to no effect on the risk of non-medically attended infections in adults.

[1] Department of Ecology and Evolution, University of Chicago, 1101 E 57th Street, Chicago, IL 60637, USA. [2] WHO Collaborating Centre for Infectious Disease Epidemiology and Control, School of Public Health, The University of Hong Kong, Patrick Manson Building, 7 Sassoon Road, Pokfulam, Hong Kong, China. These authors contributed equally: Benjamin J. Cowling, Sarah Cobey. Correspondence and requests for materials should be addressed to S. R. (email: slr@uchicago.edu)

Like many antigenically variable pathogens, influenza viruses continuously evolve to escape host immunity. As a consequence, they cause frequent epidemics and infect people repeatedly during their lives. The details of these processes—which are vital to influenza epidemiology, evolution, and the design of effective vaccines—have nonetheless remained surprisingly difficult to pin down despite nearly 70 years of study.

A major challenge is uncertainty about the nature of acquired immunity. Antibodies are the primary means of protection against influenza and impose strong selection on its surface proteins[1,2]. Antibody responses to influenza are highly cross-reactive, in that antibodies induced by infection or vaccination with one strain often protect against infections with related strains[3,4]. The hierarchical nature of this cross-reactivity, in which memory responses to conserved antigens tend to dominate over responses to new epitopes, is known as original antigenic sin[5–7]. It might underlie the phenomenon of imprinting, in which primary infection with one influenza subtype protects against severe disease and death with other subtypes that are closely related phylogenetically[8]. But the specificity and duration of protection after seasonal influenza infection have been difficult to estimate, partly because the relationship between antibody titer and protection appears complex, and also because longitudinal observations of antibody titers and infections are rare. The most common measure of anti-influenza antibody is the hemagglutination inhibition (HI) assay, and HI antibody titers are an established correlate of protection[9]. The HI titer corresponding to 50% protection against infection, commonly cited as 40[10,11], may vary by influenza A subtype and host age[12,13], although measurement error, long intervals between titer measurements, and variable titer changes after infection complicate inferences. Recent models have made progress by incorporating measurement error[14,15], representing infections as latent states[14,16,17], and using titers to historic strains to measure the intervals between infections[14], attack rates[15,16], and the breadth of the response over time[14,17]. But the relatively short periods of observation in these studies have made it difficult to estimate some basic quantities in the response to infection, namely, how long protection lasts, and whether antibody titers adequately reflect the strength of protection against infection in individuals over time.

Longitudinal cohorts provide relatively unobscured observations of the dynamics of infection and protection, and mechanistic models allow hypotheses about these dynamics to be tested. We fit stochastic mechanistic models to influenza antibody titers collected over five years from a large household cohort study including children and adults. These models account for pre-existing immunity, variation in the response to infection, and the possibility that the HI titer is not a good correlate of protection from infection. Their flexibility allows many previous assumptions to be relaxed. For both influenza A subtypes, we estimate the duration of within-subtype and cross-subtype protection, the relationship between HI titer and protection, and the effect of early childhood influenza exposures on infection risk later in life. The dynamics inferred from these individual-level models are remarkably consistent with the epidemiological dynamics of the larger population, and they also support immunological theory of how the antibody response to influenza changes with age.

## Results

**Homosubtypic protection: correlation with anti-HA antibodies.** We fitted models to data from a cohort of 592 adults (>15 y) and 114 children (≤15 y) followed from 2009 to 2014 in Hong Kong. Members of this cohort were part of a larger household study[18,19] and were selected because they were not vaccinated for the study and reported no vaccination during the

five years of follow-up. The cohort included 337 households with a median size of 2 members. Sera were obtained every six months and tested for antibodies to circulating strains of influenza A (H3N2) and A(H1N1)pdm09 via the HI assay.

Antibodies measured by the HI assay are an established correlate of protection for influenza virus infection[10,20,21]. Neutralizing antibodies against the dominant surface proteins, hemagglutinin (HA) and neuraminidase (NA), can target different sites on them, and the specificity of the antibody response appears to change with immune history and age[22–26]. HI assays measure antibodies to HA but not NA, and they disproportionately measure anti-HA antibodies that attach near the receptor binding site toward the top of the HA globular domain.

To characterize the role of these antibodies in protection, we tested a simple hypothesis about the dynamics of susceptibility after infection: protection from infection could be associated with HI titer, the time since last infection (a potential correlate of other antibody and broader immune responses), or a mixture of the two. We define an individual's susceptibility to a subtype as the probability of infection given exposure. Rewriting the hypotheses mathematically, we propose the susceptibility of an individual $i$ to subtype $s$ at time $t$, $q_{i,s}(t)$, is a function of HI-correlated factors and non-HI-correlated factors.

HI-correlated factors: An individual's susceptibility can be measured by the HI titer to a representative circulating strain. We assume that HI-correlated susceptibility, $q_{1_{i,s}}(t)$, is a logistic function of the current titer[10,11], with the shape of the curve set by the titer at which 50% of subjects are protected from infection (Fig. 1b). This 50% protective titer is defined for each age group $a \in$ {child, adult}, $\text{TP50}_{a_i,s}$ (Eq. 9),

$$q_{i,s}(t) = q_{1_{i,s}}(t). \tag{1}$$

Non-HI-correlated factors: Susceptibility can be explained by the time since last infection with that subtype (Fig. 1b). Susceptibility determined by non-HI-correlated protection, $q_{2_{i,s}}(t)$, is a function that starts at 0 (no susceptibility) immediately after infection. The susceptibility increases as protection wanes exponentially at rate $w_{\text{nonspecific},a_i,s}$ (Eq. 10),

$$q_{i,s}(t) = q_{2_{i,s}}(t). \tag{2}$$

Titers in this model are still informative as indicators of infections, but they do not affect infection risk.

We evaluate the contribution of each component by fitting a weighted susceptibility model,

$$q_{i,s}(t) = q_{1_{i,s}}(t)\psi_{a_i,s} + q_{2_{i,s}}(t)(1 - \psi_{a_i,s}), \tag{3}$$

where $\psi_{a_i,s}$ measures the contribution of HI-correlated protection to susceptibility in children ($a_i$ = children) and adults ($a_i$ = adults). The value of $\psi_{a_i,s}$ therefore distinguishes between models in which protection is completely HI-correlated ($\psi_{a_i,s} = 1$), models in which protection is completely non-HI-correlated ($\psi_{a_i,s} = 0$), and models in which protection is predicted by a combination of the two components ($0 < \psi_{a_i,s} < 1$). To estimate the contribution of HA-head-directed antibodies to protection from influenza infection in children and adults, Eq. 3 was incorporated into a partially observed Markov model that simulates individuals' latent (unobserved) HI titers and susceptibility to infection over time while simultaneously accounting for measurement error (Fig. 1b; "Methods" section). The model assumes that infection can change the antibody titer, which allows infection events and thereby latent susceptibility ($q_{i,s}(t)$) to be inferred from longitudinal sera.

In the model, infection acutely boosts an individual's titer, which then wanes slowly over one year, potentially leaving a long-

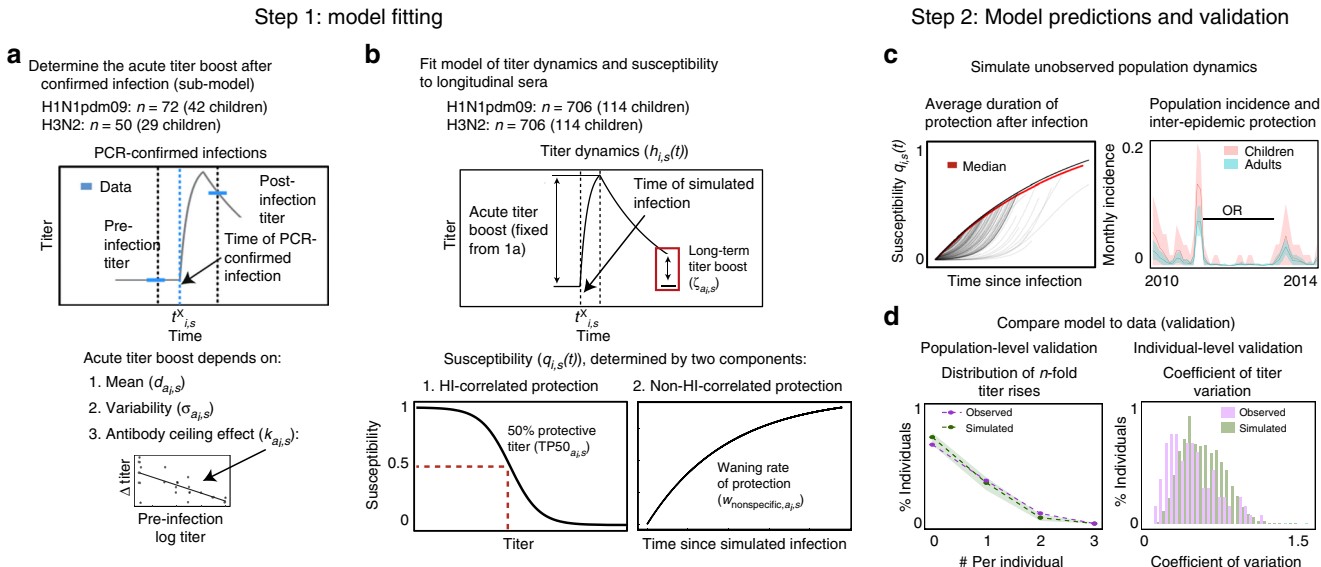

**Fig. 1** Schematic of modeling approach. Steps 1 and 2 are performed for each subtype. Step 1: Model fitting. **a** First, the sub-model of short-term post-infection titer dynamics is fitted to a subset of the data. This subset includes the time of PCR-confirmed infection and the immediate pre- and post-infection titers. The mean and standard deviation of the acute titer boost and the mean's dependence on pre-infection titer (the antibody ceiling effect) are fitted. **b** Next, fixing the parameters associated with short-term titer dynamics (**a**), the full model is fitted to titers from the entire cohort. The contribution of HI-correlated and non-HI-correlated protection, the titer waning rate, the 50% protective titer, and the long-term boost after infection are estimated. Step 2: Model predictions and validation. **c** The duration of protection and inter-epidemic protection are estimated from simulating population-level dynamics from the best-fit model in **b**. From the latent infections and susceptibility for each individual, we track the loss of protection after infection. We also estimate the cumulative epidemic incidence and the odds ratios (OR) of protection between epidemics. **d** Simulation enables additional checks of the model. We compare the simulated and observed distributions of *n*-fold titer rises and coefficients of titer variation among individuals

| Table 1 Maximum likelihood estimates and 95% confidence intervals (CI) | | | |
|---|---|---|---|
| **Subtype** | **Parameter** | | **MLE [95% CI]** |
| H3N2 | Long-term boost | $\zeta_{adults,s}$ | 0.0 [0.0, 0.03] |
| | | $\zeta_{children,s}$ | 0.04 [0.02, 0.07] |
| | Weight of HI-correlated immunity | $\psi_{adults,s}$ | 0.0 [0.0, 0.2] |
| | | $\psi_{children,s}$ | 1.0 [0.8, 1.0] |
| | 50% protective titer | $TP50_{children,s}$ | 44 [29,74] |
| | Half-life non-HI-correlated immunity | from $w_{nonspecific,\ adults,s}$ | 4.1y [3.2, 5.5] |
| | Daily within-household transmission rate | $\omega_s$ | 0.05 [0.02, 0.11] |
| H1N1pdm09 | Long-term boost | $\zeta_{adults,s}$ | 0.06 [0.05, 0.07] |
| | | $\zeta_{children,s}$ | 0.04 [0.02, 0.07] |
| | Weight of HI-correlated immunity | $\psi_{adults,s}$ | 0.10 [0.07, 0.12] |
| | | $\psi_{children,s}$ | 1.0 [0.8, 1.0] |
| | 50% protective titer | $TP50_{adults,s}$ | 8 [1,12] |
| | | $TP50_{children,s}$ | 15 [8,25] |
| | Half-life non-HI-correlated immunity | From $w_{nonspecific,\ adults,s}$ | 4.0y [3.1, 5.2] |
| | Daily within-household transmission rate | $\omega_s$ | 0.06 [0.03, 0.09] |

term boost that does not wane. To increase accuracy in modeling these acute boosts, we took advantage of 112 PCR-confirmed infections and pre- and post-infection titers from this study to fit the mean and standard deviation of the titer rises (Fig. 1a; Supplementary Discussion). The acute boost was higher for H3N2 than for H1N1pdm09 in both children and adults, but there was no significant difference in boost sizes by age in either subtype (Supplementary Table 5). We found evidence of an antibody ceiling effect, whereby individuals with higher pre-infection titers have smaller boosts (see Supplementary Discussion). After fitting this sub-model to describe the relationship between infection and short-term titer changes, we then fixed its parameters to fit the full model of titer dynamics to all 706 individuals. For children and adults, the full model estimates the contribution of HI-correlated and non-HI-correlated factors to

protection (Eq. 3), the magnitude of the long-term titer boost, the 50% protective titers (for Eqs. 1 and 3), and the rate of waning of non-HI-correlated protection (for Eqs. 2 and 3) (Fig. 1b). Additionally, because some subjects in the study belong to the same household, we estimate the contribution of infected household members to an individual's force of infection, relative to that of the community. Simulating from the maximum likelihood estimates of the best model yields additional information, including the typical duration of protection after infection, attack rates in different epidemics, and the odds ratios of infection from one epidemic to the next (Fig. 1c). These simulations are also useful for checking how well the model reproduces different features of the data (Fig. 1d).

For both subtypes, protection in children is HI-correlated ($\psi_{children,H3N2} = 1.0$, 95% CI: (0.8, 1.0); $\psi_{children,H1N1pdm09} = 1.0$,

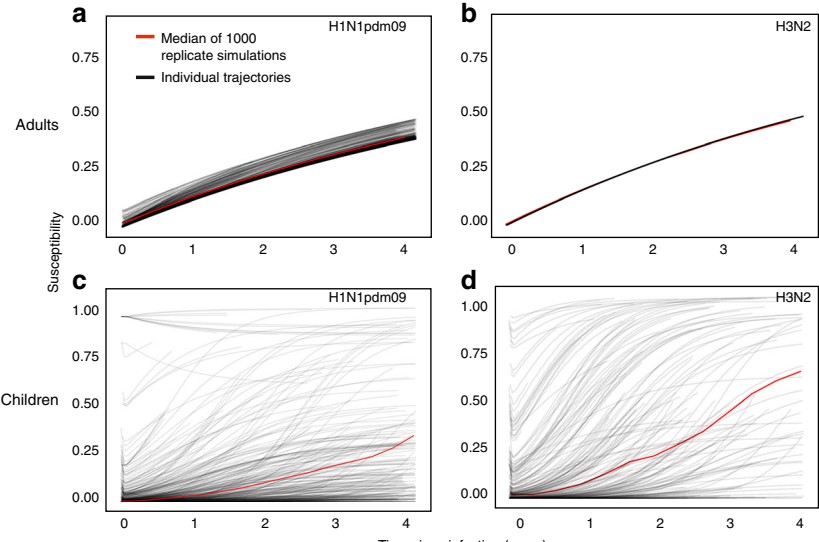

**Fig. 2** Susceptibility after simulated infection with H1N1pdm09 and H3N2 in adults and children. Susceptibility is shown after simulated infection (at time $t = 0$) for adults with H1N1pdm09 (**a**) and H3N2 (**b**) and for children with H1N1pdm09 (**c**) and H3N2 (**d**). The black lines represent individual trajectories from one simulation, and the red line represents the median among individuals over 1000 replicate simulations. Curves from individual trajectories are truncated at points corresponding to the end of the study

95% CI: (0.8, 1.0); Table 1), whereas in adults, time since infection better predicts protection ($\psi_{\text{adults,H3N2}} = 0.0$, 95% CI (0.0, 0.2); $\psi_{\text{adults,H1N1pdm09}} = 0.1$, 95% CI (0.07, 0.12)). This result suggests that early in life, protection against influenza virus infection is dominated by immune responses that correlate well with HI titer, such as antibodies to the top of the HA head. However, over time, other immune responses dominate, such that time since infection becomes a better predictor of protection than HI titer. This result is consistent with the observation that more children than adults in this study have detectable baseline HI titers, and children have higher mean baseline HI titers, to circulating strains (Supplementary Fig. 1; "Methods" section).

**Duration of age- and subtype-specific protection.** Using the best-fit models for each subtype, we next quantified the duration of protection against infection with the same subtype in adults and children.

To estimate the duration of protection in adults, we simulated from the fitted models for each subtype. Using 1000 replicate simulations, we tracked the latent susceptibility after infection. For each individual at any time, this susceptibility is given by the weighted average of two components, one set by the titer and the other by the time since infection (Eq. 3). In adults, the model favors non-HI-correlated protection against H3N2, with a half-life of 4.1 y (95% CI (3.2, 5.5)) (Table 1).

In contrast to H3N2, however, HI-correlated protection contributes slightly to adults' protection against H1N1pdm09 ($\psi_{\text{adults,H1N1pdm09}} = 0.1$ 95% CI (0.07, 0.12)). The associated low 50% protective titer (TP50$_{\text{adults,H1N1pdm09}} = 8$ 95% CI (1,12), Supplementary Fig. 2) implies that most individuals (i.e., even individuals with titer <10, the lower limit of detection by HI assays) benefit from minor additional protection after infection. When we plot the latent susceptibility after each infection with each subtype, we see that the individual trajectories in Fig. 2a, b follow the curve describing the change in non-HI-correlated protection (Eq. 10). The contribution of HI-correlated protection to the post-infection protection dynamics of H1N1pdm09 creates mild individual-level variability arising from differences in pre-

infection titers. In aggregate, adults' susceptibility to H1N1pdm09 thus wanes with a half-life of approximately 6.4 y (95% quantile: (4.8, 6.7)), reflecting a significantly slower loss of protection than for H3N2. Infection in adults produces only a small durable titer boost to H1N1pdm09 ($\zeta_{\text{adults,H1N1pdm09}} = 0.06$ (95% CI: 0.05, 0.07)) and a negligible durable boost to H3N2 ($\zeta_{\text{adults,H3N2}} = 0.0$, (95% CI: 0.0, 0.03); Table 1). Compared to adults, children have a more variable duration of protection. Because susceptibility in children depends only on HI titer, the dynamics of individual protection are sensitive to pre-infection titers and differences in the magnitude of the acute boost post-infection. For both H1N1pdm09 and H3N2, we estimated substantial variation in the short-term titer dynamics after PCR-confirmed infection (see Supplementary Discussion). The variability arises both from stochastic variation in the magnitude of the short-term titer boost and from the antibody ceiling effect (Supplementary Tables 1 and 5). Protection in children wanes with a median half-life of ~7.1 y (95% quantile: (2.8, 8.8)) for H1N1pdm09 and 3.5 y (95% quantile: (1.4, 5.2)) for H3N2 (Fig. 2c, d); thus, the duration of protection in children is similar to adults' against H1N1pdm09, but shorter against H3N2.

We find that unlike in adults, infection with H1N1pdm09 generates a long-term boost in titer that is 30% the size of the acute boost ($\zeta_{\text{children},s} = 0.3$ (95% CI: 0.2, 0.5), Table 1), allowing children to gain long-term protection as their baseline titer eventually rises above the TP50$_{\text{children}}$ through repeated exposures. In H3N2, by contrast, we estimate only a small long-term boost ($\zeta_{\text{children},s} = 0.04$ (95% CI: 0.02, 0.07), Table 1), which could reflect the antigenic evolution of circulating strains and the change in the strain used in the HI assay during the study.

**Population-level estimates of incidence and protection.** Despite being fitted to individuals' titers, the models recover reasonable population-level patterns of infection for both subtypes. From the simulated latent infections, we inferred the annual incidence and the cumulative epidemic incidence in children and adults (Fig. 3, Supplementary Table 2). Because the models assume that the community-level, subtype-specific influenza intensity affects an

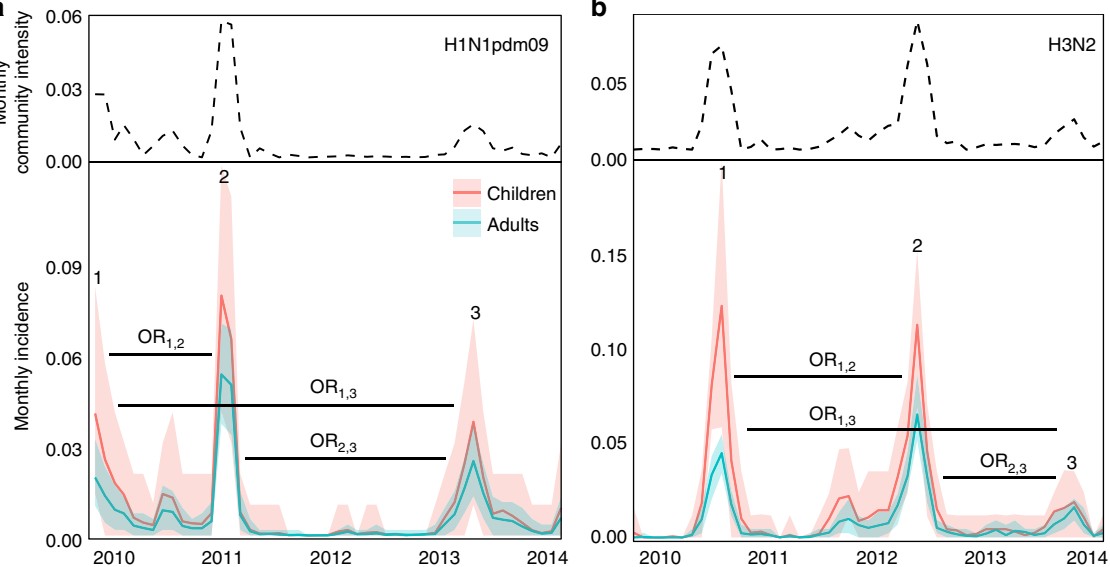

**Fig. 3** Simulated monthly H1N1pdm09 and H3N2 incidence. Simulated monthly incidence for H1N1pdm09 (**a**, bottom) and H3N2 (**b**, bottom) in children and adults, averaged over 1000 simulations, contrasted with respective monthly community intensities (**a**, **b**, top). The shaded areas are bounded by the 2.5 and 97.5% quantiles from the simulations. Horizontal black bars denote inter-epidemic periods for odds ratios (OR)

| Table 2 Inter-epidemic odds ratios of infection | | | | |
|---|---|---|---|---|
| **Subtype** | | | **OR [95% quantiles]** | **Estimate from [19]** |
| H1N1pdm09 | Adults | $OR_{1,2}$ | 0.55 [0.51, 0.70] | 0.27 [0.10, 0.76] |
| | | $OR_{2,3}$ | 0.72 [0.69, 0.80] | |
| | | $OR_{1,3}$ | 0.86 [0.74, 1.16] | |
| | Children | $OR_{1,2}$ | 0.39 [0.12, 0.85] | |
| | | $OR_{2,3}$ | 0.29 [0.08, 0.65] | |
| | | $OR_{1,3}$ | 0.48 [0.09, 1.10] | |
| H3N2 | Adults | $OR_{1,2}$ | 0.37 [0.19, 0.56] | 0.39 [0.18, 0.83] |
| | | $OR_{2,3}$ | 0.55 [0.38, 0.76] | |
| | | $OR_{1,3}$ | 0.81 [0.53, 1.03] | |
| | Children | $OR_{1,2}$ | 0.48 [0.10, 0.89] | |
| | | $OR_{2,3}$ | 0.28 [0.06, 0.79] | |
| | | $OR_{1,3}$ | 0.85 [0.82, 1.09] | |

Odds ratios are predicted from 1000 replicate simulations of the models for H1N1pdm09 and H3N2 at the MLEs. Epidemics are numbered as in Fig. 3. The rightmost column shows odds ratios estimated using ≥4-fold changes in titer as an indicator of infection

individual's infection risk (Methods, Eq. 7), it is unsurprising that periods of high monthly incidence in the simulated study population match those in the community (Fig. 3). However, the absolute monthly incidence in the study population is effectively unconstrained, emerging from the estimated subtype-specific scaled transmission rate, $\beta_{\text{scaled},s}$ (Supplementary Table 7, Supplementary Fig. 21), and protection parameters (Table 1). The results nonetheless match estimates from other populations. The cumulative incidences of individual H1N1pdm09 epidemics range from 7 to 12% in adults and 10–17% in children (Supplementary Table 2). For H3N2, the cumulative epidemic incidences range from 4–12% in adults and 6–23% in children. Estimates of cumulative seasonal influenza incidence in the United States are 5–20% based on combined serology and viral infection (of influenza A and B)[27] and 3–11% based on symptomatic PCR-confirmed infections of influenza A[28]. Converted to annual rates, the simulated annual incidences are similar, ranging from 5–17% in adults and 7–29% in children. We estimate that for both subtypes, the daily within-household intensity (the increased risk of transmission given that at least one member in the household

is infected) is roughly 5%, or ~25% over an average five-day infectious period ($\omega_s$, Table 1). This is consistent with published estimates of secondary household attack rates for influenza A virus ranging from 10 to 30%[29–34]. For example, for a young child at the peak of a H1N1pdm09 epidemic, the daily risk of transmission from the community is approximately 2.1% per day. Therefore, our results suggest that the within-household transmission rate is at least twice as high as the maximum community-level risk. The same is true for H3N2.

The simulated infections reproduce other estimates of protection over time. We estimated the odds ratios of protection between epidemics (Table 2). We find evidence of inter-epidemic H1N1pdm09 protection for children between 2009 and 2011, consistent with a previous analysis of this trial that used ≥4-fold titer rises to indicate infection[19], and between 2011 and 2013. We also find evidence of protection for adults for the same two inter-epidemic periods. Protection against H3N2 in both children and adults occurred between 2010 and 2012 and between 2012 and 2013. The point estimates of the odds ratios inferred from ≥4-fold titer rises in children are lower than those inferred from latent

infections in this model, suggesting that past infection may protect more against large titer rises than infection per se.

**Group-level HA imprinting and heterosubtypic protection.** Previous work has suggested that primary infection with a subtype reduces susceptibility to severe disease and death with related subtypes, a phenomenon known as imprinting[8,35]. Influenza A HAs fall into two phylogenetic groups, with H1 and H2 belonging to Group 1 and H3 to Group 2. We estimated the protective effect $\alpha_{\text{imp},s}$ of primary infection with a subtype of one HA group on the rate of infection with subtype $s$ of the same HA group by

$$\lambda_{\text{imp},i,s}(t) = \lambda_{i,s}(t)(\alpha_{\text{imp},s})(p_{\text{imp},i,s}) + \lambda_{i,s}(t)(1 - p_{\text{imp},i,s}), \quad (4)$$

where $\lambda_{\text{imp},i,s}(t)$ is the force of infection on individual $i$ with subtype $s$ at time $t$ considering imprinting, and $\lambda_{i,s}(t)$ is the baseline infection rate for an individual not imprinted with that subtype's HA group (Eq. 6). The probability of having had a primary infection with the same group as subtype $s$ is $p_{\text{imp},i,s}$, and $\alpha_{\text{imp},s}$ is thus the change in that baseline force of infection from imprinting. Imprinting implies $\alpha_{\text{imp},s} < 1$. We calculate $p_{\text{imp},i,s}$ based on the individual's birth date, the current date, and historical incidence data ("Methods" section, Supplementary Fig. 3A). Birth-year effects and age-specific effects are often confounded, but are potentially distinguishable in longitudinal data from individuals of similar ages but different primary exposures. We therefore fit the imprinting models for H1N1pdm09 and H3N2 to data from middle-aged adults (35–50 y), whose first exposures were to Group 1 (mainly H2N2) or Group 2 (H3N2) viruses (Supplementary Fig. 3B). For H3N2, we thus estimate the effect of homosubtypic imprinting, and for H1N1pdm09, we estimate group-level imprinting from primary infection with either H1N1 or H2N2 (Supplementary Table 3). The 95% confidence intervals for the maximum likelihood estimates of the imprinting effect (Supplementary Fig. 3C) show the model is consistent with Group 1 protection ranging from 0.6 to 1.0, i.e., 0–40% reduction in susceptibility, and with Group 2 protection between 0.9 and 1.2, suggesting no protection.

Epidemiological and immunological studies have suggested that infection with one subtype might protect in the short term against another[36–38]. To estimate the duration of heterosubtypic protection, we fitted a two-subtype model of H1N1pdm09 and H3N2, fixing the parameters that govern homosubtypic immunity at the MLEs of the best-fit single-subtype models (Table 1). Let $q_{\text{homosubtypic},i,s}$ denote susceptibility to subtype $s$ determined only by homosubtypic protection. Heterosubtypic protection after infection with subtype $m \neq s$ contributes to the susceptibility against subtype $s$ such that the net susceptibility to subtype $s$, $q_{i,s}(t)$, is

$$q_{i,s}(t) = \min(q_{\text{hetero},i,s}(t), q_{\text{homosubtypic},i,s}(t)), \quad (5)$$

where $q_{\text{hetero},i,s}(t)$ is determined by the time since infection with subtype $m$ (Eq. 17). We assumed the rate of waning of heterosubtypic protection, $w_{\text{nonspecific},m}$, is identical for both subtypes. In these data, the model estimates that any heterosubtypic protection is fleeting (Supplementary Fig. 4; half-life from $w_{\text{nonspecific},m} = 0.002$ y; 95% CI: 0.0, 0.07).

**Model validation and sensitivity analysis.** In addition to comparing the models' results to other estimates of population-level incidence and protection between epidemics, we investigated the model's ability to match other features of the data. The best-fit models reproduce the observed distributions of 1−, 2−, and 4-fold titer rises, considering all individuals' trajectories together (Supplementary Discussion; Supplementary Figs. 6 and 7).

However, the models tend to overestimate how much an individual's titer varies over time (Supplementary Discussion; Supplementary Figs. 8 and 9). This suggests the model might not be fully capturing individual heterogeneity in infection risk and/or the response to infection, although individuals' trajectories appear reasonable by eye (Supplementary Figs. 10, 11). Beyond estimating the factors that correlate with protection, we examined the robustness of our model to other assumptions. Our results are robust to changes in the initial conditions, namely, how recently individuals are assumed to have been infected (see Supplementary Discussion). Results also do not change with an alternate scaling of the community influenza intensity to account for increased surveillance during the 2009 H1N1pdm09 pandemic (see Supplementary Discussion, Supplementary Figs. 12 and 19). Additionally, our assumptions about the m]easurement error are consistent not only with values estimated by others[17,39] but also with the error estimated from replicate titer measurements in the data (see Supplementary Discussion, Supplementary Fig. 20).

## Discussion

Our results suggest that protection against influenza A has different origins in adults and children. In children, the HI titer is a good correlate of protection, and infection durably boosts titers against H1N1pdm09. In adults, time since infection is more strongly correlated with protection than the HI titer, and infection is associated with small to no long-term changes in titer. HI assays primarily measure antibodies to epitopes on the top of HA, and not epitopes on the sides of HA or on the stalk. Thus, these results suggest that children tend to produce antibodies that target the head of the HA, which could mediate protection, whereas adults rely on antibody responses to other sites or potentially other forms of immunity for protection. This model is consistent with the concepts of antigenic seniority and original antigenic sin. Antigenic seniority refers to the phenomenon that individuals' highest antibody titers to influenza are to strains that circulated in childhood[22], and original antigenic sin is the process by which antibody responses to familiar sites are preferentially reactivated on exposure to new strains[5–7]. With time, these familiar sites may be the ones that are most conserved. On HA, these sites would tend to be away from the fast-evolving epitopes near the receptor binding domain, and would not be readily detected by HI assays. Consistent with this view, epidemiological studies have shown that levels of stalk-directed antibodies increase with age[40,41], and that vaccination with a partially antigenically novel influenza virus boosts responses to conserved sites, including the stalk[42–45]. Our model shows that these differences are persistent and relate to protection. In adults, the time since last infection better correlates with protection than HI titer because their antibodies to the top of the HA head tend to be antigenically mismatched. Compared to children, more of their protection derives from other responses.

We estimated that protection in both children and adults wanes with an average half-life of 3.5–7 years, lasts longer against H1N1pdm09 than H3N2, and lasts slightly longer in adults compared to children against H3N2. These timescales are consistent with the estimated decay of immunity over 2–10 years due to antigenic evolution in population-level models[46,47]. Responses may be more durable to H1N1pdm09 compared to H3N2, and in adults compared to children for H3N2, because the epitopes that are targeted are relatively more conserved. In contrast to adults, the dependence of protection on HI titer in children leads to substantial variation in susceptibility over time (Fig. 2). This heterogeneity may well extend to adults but is difficult to identify without much longer time series or other immune assays. The models' tendency to overestimate individuals' titer variation over

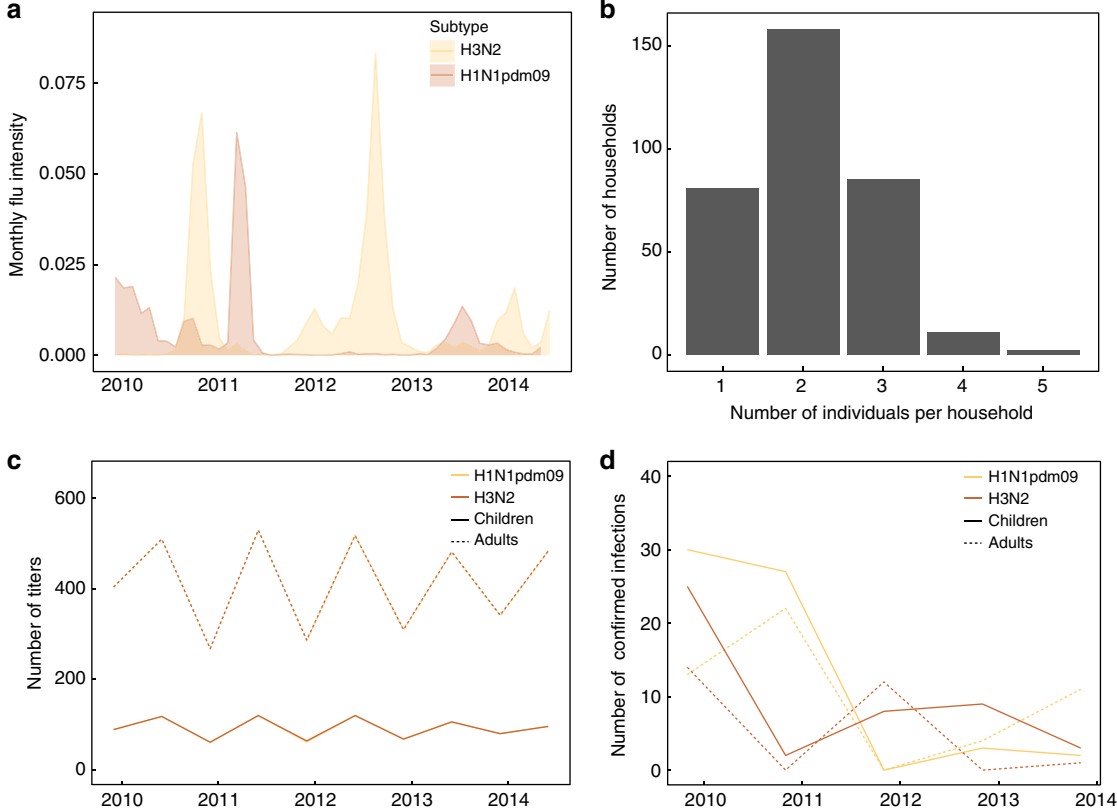

**Fig. 4** Features of the data. **a** Community intensity of H3N2 and H1N1pdm09 (ILI × % influenza-positive) over the study. **b** Distribution of household sizes in the study cohort. **c** Number of titer samples for H3N2 and H1N1pdm09 in children and adults over the study. **d** Number of PCR-confirmed infections with H3N2 and H1N1pdm09 in children and adults over the study. The raw data are provided as a Data Source file

time suggests that important differences between individual responses could be missing (Supplementary Figs. 8 and 9). Longer observation periods and more complete observations of the immune response can help separate these factors from behavioral or environmental differences in infection risk.

We find minimal evidence that HA imprinting and heterosubtypic immunity affect susceptibility to infection. Previous analyses of HA group-level imprinting have suggested that imprinting reduces the rate of severe disease and death[8]. Serological testing in this study occurred independent of symptoms (see Supplementary Discussion, Supplementary Table 6, Supplementary Fig. 16). If the model were estimating protection from symptomatic, medically attended infections instead, a stronger effect might have been supported. In the same vein, heterosubtypic immunity, for which there is good evidence[36], might reduce the severity of illness rather than prevent infection[48,49]. Another possibility is that the discordance of H1N1pdm09 and H3N2 epidemic peaks in this study (Fig. 3 and Supplementary Fig. 5) reduced the model's power to detect short-term cross-protection (Supplementary Fig. 4).

This work has several limitations. We return again to the concept of heterogeneity. Though our models support substantial variability in the short-term titer boost after infection, our data lack multiple PCR-confirmed infections from the same people. Thus, we cannot distinguish the nonspecific variability at each infection ($\sigma_{a_i,s}$) from consistent differences between individuals, which might be expected if people persistently target different sites on HA, NA, and other proteins. However, the statistically indistinguishable acute titer boosts in children and adults indicate that despite adults' low baseline HI titers, we were not systematically underestimating their rate of infection. Although our results provide insight into differences between children and

adults, we have obviously not modeled the evolving response in individuals over a lifetime, spanning infancy to old age. We thus cannot resolve how age-related phenomena interact with immune history to affect the response to infection.

Broadly, our results estimate several years of protection after infection with influenza A in children and adults, and suggest that this protection is associated with different immune responses, which are consistent original antigenic sin. These results underscore the need for a deeper understanding of the factors that determine the variable response to infection among individuals, and for better correlates of immune protection. They also underscore the utility of longitudinal cohorts and mechanistic models to investigate the dynamics of influenza.

## Methods
**Study description**. The data are part of a community-based study of influenza virus infections in households that was conducted in Hong Kong between 2009 and 2014 (clinical trial NCT00792051).[19] The study tracked individuals in 796 households, each of which included at least 1 child aged 6–17 y that had no contraindications against the trivalent inactivated influenza vaccine (TIV). One eligible child 6–17 y of age per household was randomized to receive either a single dose of TIV or saline placebo, regardless of influenza vaccination history. In vaccinated individuals, sera was collected at baseline prior to vaccination (August 2009 —February 2010) and 1 month after vaccination. In all individuals, sera was collected after enrollment in the autumn of 2009 and again each subsequent autumn, and each spring for at least 25% of participants. Participants were invited annually to continue enrollment. Individuals reported receipt of the influenza vaccine outside of the trial annually.

Participants and household contacts were encouraged to record systemic and respiratory symptoms daily in diaries. Acute respiratory infections (ARIs) were surveilled by telephone calls every 2 weeks, and households were encouraged to report ARIs promptly to the study hotline. Home visits were triggered by the presence of any 2 of the following: fever (≥ 37.8 ℃), chills, headache, sore throat, cough, presence of phlegm, coryza, or myalgia in any household member.

Combined nasal and throat swabs were collected from all household members during home visits, regardless of illness.

**Ethical approval.** The study protocol was approved by the Institutional Review Board of the University of Hong Kong. All adults provided written consent. Parents or legal guardians provided proxy written consent for participants ≤17 y old, with additional written assent from those 8–17 y.

**Laboratory testing.** Serum specimens were tested by HI assays in serial doubling dilutions from an initial dilution of 1:10[18,19]. The antibody titer was taken as the reciprocal of the greatest dilution that gave a positive result. Sera from year 1 (2009–2010) and year 2 (2010–2011) were tested against A/California/7/2009 (H1N1) and A/Perth/16/2009-like (H3N2). In years 3–5 (2011–2012, 2012–2013, and 2013–2014), sera were tested against the same H1N1pdm09 strain and against A/Victoria/361/2011-like (H3N2). Sera from consecutive years were tested in parallel, such that duplicate titer measurements exist for sera sampled during the middle of the study. For this analysis, we used the first titer measurement obtained for any serum sample. Nose and throat swabs were tested by reverse transcription polymerase chain reaction (PCR) for influenza A and B viruses using standard methods, as described previously[50].

**Data included in this analysis.** We fitted models to HI titers from 706 individuals (including 114 children ≤15 y old at enrollment) from 337 different households that were not vaccinated as part of the study and reported no vaccination at any season during follow-up. We excluded individuals with any missing vaccination information. Individuals in this subset were sampled at a median of 6.6 months over a median 5.0 years of follow-up. Households ranged in size from 1 to 5 members (median = 2 members). Figure 4c gives the number of samples over the study among children and adults. Among children, the median age at enrollment was 11 y, and the age range was 3–15 y. Among adults, the median age of enrollment was 43 y, the age range was 16–77 y, and 89% of adults were between 25 and 55 y. We fitted sub-models to data from 50 individuals (including 29 children ≤15 y old at enrollment) with PCR-confirmed H3N2 infection and 78 individuals (including 42 children ≤15 y old at enrollment) with PCR-confirmed H1N1pdm09 infection (see Supplementary Discussion). No individuals had multiple PCR-confirmed infections. The data for this analysis are the date of the subtype-specific PCR-positive nasal swab and the closest titer measurements surrounding the positive swab. For H3N2, the median time between the pre-infection titer measurement and the PCR-positive swab was 5.3 months, and the median time between the PCR-positive swab and the post-infection titer measurement was 2.6 months. For H1N1pdm09, the median time between the pre-infection titer measurement and the PCR-positive swab was 2.4 months, and the median time between the PCR-positive swab and the post-infection titer measurement was 6.6 months. Figure 4d shows the number of PCR-confirmed infections by subtype in adults and children over time.

**Complete model description.** The model simulates the titer and infection dynamics for each individual. Briefly, an individual's instantaneous infection rate or force of infection, $\lambda_{i,s}(t)$, is determined by the rate of exposure to infectious contacts in the community and household and by the individual's susceptibility, defined as the probability of infection per infectious contact. Infections occur stochastically and can change the latent titers. Simulating the model generates a latent titer for each individual at each observation. The measurement model then calculates the likelihood of each observed titer given the contemporaneous latent titer. The log likelihood of the model under any set of parameters is then the sum of the log likelihoods across individuals, which are the sums of the log likelihoods across observations. The titer and infection dynamics are described in this section.

Exposure to infection:

The instantaneous per capita infection rate for individual $i$ with subtype $s$ is denoted $\lambda_{i,s}(t)$. An individual's overall rate of infection with subtype $s$ comprises a rate of infection from the community, $\lambda_{\text{community},i,s}$, and another from the household, $\lambda_{\text{household},i,s}$,

$$\lambda_{i,s}(t) = \lambda_{\text{community},i,s}(t) + \lambda_{\text{household},i,s}(t). \quad (6)$$

The rate of exposure to infection in the community is influenced by age-specific contact rates[12,20], the age distribution of infectious contacts[51], and a proxy for the prevalence of the subtype in the community. The community-driven rate of infection is the rate of exposure modified by the individual's susceptibility to that subtype,

$$\lambda_{\text{community},i,s}(t) = q_{i,s}(t)\beta_{\text{scaled},s}L_s(t)\sum_{n=1}^{n=N_{\text{cat}}}\beta_{c,\text{cat}_i,\text{cat}_n}p_{\text{cat}_n} \quad (7)$$

where $q_{i,s}(t)$ is the individual's susceptibility to that subtype (or per-infectious-contact probability of infection), $\beta_{c,\text{cat}_i,\text{cat}_n}$ is the fixed contact rate for an individual of age category $\text{cat}_i$ with individuals of age category $\text{cat}_n$ (Supplementary Table 4, Supplementary Discussion), $N_{\text{cat}}$ is the number of contact age categories (5 in total, Supplementary Table 4), $L_s(t)$ is a proxy of influenza activity for subtype $s$, and $p_{\text{cat}_n}$

is the fraction of influenza infections attributable to age class $\text{cat}_n$. The parameter $\beta_{\text{scaled},s}$ scales the flu intensity to determine the per-infectious-contact transmission rate at time $t$. We calculate $L_s(t)$ from weekly community surveillance data as (ILI/total general practitioner consultations)(% specimens positive for subtype $s$) (Fig. 4)[52]. We impose a minimum threshold $\min(L_s(t)) = 10^{-5}$.

Individual $i$ also experiences infection risk from household members,

$$\lambda_{\text{household},i,s}(t) = q_{i,s}(t)\omega_s I_{\text{other},s}(t), \quad (8)$$

where $q_{i,s}(t)$ is the individual's susceptibility, and $I_{\text{other},s}(t)$ is an indicator variable that equals 1 if any other household member is infected with subtype $s$ at time $t$ and 0 if not. Because we do not track households in their entirety (we sample only individuals that were not vaccinated during the study), we do not model density-dependent within-household transmission. Rather, $\omega_s$ describes the daily influenza exposure rate that individual $i$ experiences with the presence of any infected family member.

Susceptibility to infection based on HI titer to the infecting strain, non-HI-correlated protection, or both:

Individual $i$'s susceptibility to subtype $s$ at time $t$, $q_{i,s}(t)$, is defined as the probability of infection given contact with an infected. Complete protection corresponds to $q_{i,s}(t) = 0$, complete susceptibility to $q_{i,s}(t) = 1$, and $0 < q_{i,s}(t) < 1$ to partial protection. We use two base functions to model susceptibility. One function assumes susceptibility depends on the HI titer against the infecting strain (the HI-correlated component), and the other on the time since infection with that subtype (the non-HI-correlated-component). The HI-correlated component of susceptibility, $q_{1_{i,s}}(t)$, is a logistic function of the HI titer[11,53] (Fig. 1b). Because previous studies suggest that the relationship between titer and susceptibility changes with age[13], we estimate the relationship separately for children and adults. The HI-correlated susceptibility of individual $i$ to subtype $s$ at time $t$, $q_{1_{i,s}}(t)$, is given by the logistic function,

$$q_{1_{i,s}}(t) = 1 - \frac{1}{1 + e^{\phi(\log(h_{i,s}(t)) - \log(\text{TP50}_{a_{i,s}}))}}, \quad (9)$$

where $h_{i,s}(t)$ is the latent titer and $\text{TP50}_{a_{i,s}}$ is the subtype- and age-specific 50% protective titer. The scaling parameter $\phi$, which determines the shape of the logistic curve, is fixed (Supplementary Table 4). The non-HI-correlated component of susceptibility, $q_{2_{i,s}}(t)$, assumes initially complete protection that wanes at a constant rate after infection,

$$q_{2_{i,s}}(t) = 1 - e^{-w_{\text{nonspecific},a_{i,s}}\left(t - t_{i,s}^{X}\right)}, \quad (10)$$

where $w_{\text{nonspecific},a_{i,s}}$ is the rate of waning, fitted separately for children and adults, and $t_{i,s}^{X}$ is the time of infection.

The susceptibility $q_{i,s}(t)$ is the weighted average of the two components (Eq. 3).

Titer dynamics after infection:

Infections can affect the latent titers. We model the post-infection titer dynamics as a series of equations that describe the acute boost, the waning from peak titer, and the potential long-term titer boost. Together, these equations determine the latent titer of individual $i$ against subtype $s$ at any time $t$, $h_{i,s}(t)$.

When individual $i$ is infected with subtype $s$, antibody titers increase from the time of infection and eventually peak. The acute boost occurs according to $f_{\text{rise}}$,

$$f_{\text{rise}}(h_{i,s}(t_{i,s}^{X}), t_{i,s}^{X}, t) = h_{i,s}(t_{i,s}^{X})^{(1-k_{a_{i,s}})}d_{i,s}(t_{i,s}^{X})\left(1 - e^{-r(t-t_{i,s}^{X})}\right), \quad (11)$$

where $t_{i,s}^{X}$ and $h_{i,s}(t_{i,s}^{X})$ give the time and titer, respectively, of the most recent infection; $r$ gives the fixed rate of titer rise after infection (Supplementary Table 4); and $d_{i,s}(t_{i,s}^{X})$ is the magnitude of the short-term boost. Recall that the time $t_{i,s}^{X}$ of a simulated infection is driven by the individual's infection rate, $\lambda_{i,s}(t)$. The age- and subtype-specific parameter $k_{a_{i,s}}$ determines the dependence of the titer boost on the pre-infection titer. When positive, it allows for an antibody ceiling effect[54], whereby higher pre-infection titers have smaller boosts (see Supplementary Discussion, Supplementary Tables 1 and 5, Supplementary Figs. 13–15).

Multiple studies demonstrate heterogeneity in the short-term titer rise after infection[39,55]. Therefore, we allow for variability in the magnitude of the short-term boost for each infection,

$$\log(d_{i,s}(t_{i,s}^{X})) \sim \mathcal{N}(d_{a_{i,s}}, \sigma_{a_{i,s}}), \quad (12)$$

where $d_{a_{i,s}}$ and $\sigma_{a_{i,s}}$ give the age- and subtype-specific log mean and standard deviation, respectively, of the boost. We estimate the parameters $k_{a_{i,s}}$, $d_{a_{i,s}}$, and $\sigma_{a_{i,s}}$ from a sub-model fitted to data from individuals with PCR-confirmed infection (see Supplementary Discussion). We then fix the values of these parameters in the main model.

After peaking at time $t_{i,s}^{P}$, the titer wanes exponentially at a fixed rate $w$ (Supplementary Table 4) to an individual's subtype-specific baseline titer,

$h_{\text{baseline},i,s}(t)$. Therefore, the titer after the peak short-term response is given by

$$f_{\text{wane}}(h_{i,s}(t_{i,s}^{\text{P}}), t_{i,s}^{\text{P}}, t) = (h_{i,s}(t_{i,s}^{\text{P}}) - h_{\text{baseline},i,s}(t))e^{-w(t-t_{i,s}^{\text{P}})}. \quad (13)$$

Infection may cause a long-term boost, $d_{\text{longterm},i,s}, (t_{i,s}^{\text{X}})$, that does not wane, where $d_{\text{longterm},i,s}(t_{i,s}^{\text{X}})$ is defined as a fraction $\zeta_{a_i,s}$ of the acute boost,

$$d_{\text{longterm},i,s}(t_{i,s}^{\text{X}}) = \zeta_{a_i,s} d_{i,s}(t_{i,s}^{\text{X}}). \quad (14)$$

The long-term boost changes the baseline titer after each infection at time $t_{i,s}^{\text{X}}$,

$$h_{\text{baseline},i,s}(t) = h_{\text{baseline},i,s}(t_{i,s}^{\text{X}}) + d_{\text{longterm},i,s}(t_{i,s}^{\text{X}}). \quad (15)$$

Let $T_{\text{peak}}$ denote the fixed length of time between infection and peak titer (Supplementary Table 4). The complete expression for $h_{i,s}(t)$ is then

$$\begin{cases} h_{i,s}(t) = h_{i,s}(t_{i,s}^{\text{X}}) + f_{\text{rise}}(h_{i,s}(t_{i,s}^{\text{X}}), t_{i,s}^{\text{X}}, t), & \text{for } t - t_{i,s}^{\text{X}} < T_{\text{peak}}, \\ h_{i,s}(t) = h_{\text{baseline},i,s}(t_{i,s}^{\text{X}}) + f_{\text{wane}}(h_{i,s}(t_{i,s}^{\text{P}}), t_{i,s}^{\text{P}}, t), & \text{for } t - t_{i,s}^{\text{X}} \geq T_{\text{peak}}. \end{cases} \quad (16)$$

Heterosubtypic immunity:

Heterosubtypic immunity acts as a nonspecific form of protection against subtype $s$ following infection with subtype $m$ at time $t_{i,m}^{\text{X}}$ and wanes at rate $w_{\text{nonspecific},m}$,

$$q_{\text{hetero},i,s}(t) = 1 - e^{-w_{\text{nonspecific},m}(t-t_{i,m}^{\text{X}})}. \quad (17)$$

**Measurement model and likelihood function.** The observed titer $H_{\text{obs},i,s}(t)$ relates to the corresponding latent titer $h_{i,s}(t)$ via the measurement model, which accounts for error in the titer measurements and the effect of discretization of titer data into fold-dilutions. The observed titers are fold-dilutions in the range [<1:10, 1:10, 1:20…, 1:5120]. Consistent with other models[12,14,16], we define a log titer (log $H$) for any titer $h$,

$$\log H = \log_2\left(\frac{h}{10}\right) + 2, \quad (18)$$

such that the log of the observed titer takes on discrete values in the range[1,11]. In order to relate the observed titer, $H_{\text{obs},i,s}(t)$, to the latent titer, $h_{i,s}(t)$, we begin by transforming both into log titers (Eq. 18), yielding the log observed titer $\log H_{\text{obs},i,s}(t)$ and the log latent titer $\log H_{i,s}(t)$ We assume that the observed log titer to subtype $s$ is normally distributed around the log latent titer,

$$\log H_{\text{obs},i,s}(t) \sim \mathcal{N}(\log H_{i,s}(t), \epsilon), \quad (19)$$

where $\varepsilon$ gives the standard deviation of the measurement error.

Following other analyses that quantified the measurement error associated with different titers[15,39], we assign a lower measurement error ($\varepsilon = 0.74$ log titer units, Supplementary Table 4) for undetectable (<10) titers. The observed titer is censored at integer cutoffs, such that the likelihood of observing $\log H_{\text{obs},i,s}(t) = j$ given log latent titer $\log H_{i,s}(t)$ is

$$\mathcal{L}(j \mid \theta, \log H_{i,s}(t)) = \begin{cases} f(\log H_{i,s}(t) \leq j), & j = 1 \\ f(j \leq \log H_{i,s}(t) \leq j+1), & 2 \leq j \leq 10 \\ f(\log H_{i,s}(t) \geq j), & j = 11 \end{cases} \quad (20)$$

where $\theta$ gives the parameter vector and $f$ is specified as in Eq. 19. For each individual, the log likelihood is the sum of the log likelihoods across observations,

$$\mathcal{L}_i = \Sigma_{n=1}^{n=n_{\text{obs}}} \mathcal{L}(j \mid \theta, \log H_{i,s}(t_n)), \quad (21)$$

where $n_{\text{obs}}$ is the number of observations for individual $i$ and $t_n$ gives the time at observation $n$. The log likelihood of the model for any parameter set $\theta$ is then the sum across $n_{\text{ind}}$ individuals,

$$\mathcal{L} = \Sigma_{i=1}^{n_{\text{ind}}} \mathcal{L}_i. \quad (22)$$

Supplementary Table 4 summarizes the model parameters and state variables.

**Initial conditions.** We assign each individual's initial latent subtype-specific baseline titer, $h_{\text{baseline},i,s}(0)$, based on that individual's lowest observed titer during the study, $h_{\text{obs},i,s}^{\min}$. Because an observed HI titer represents the lower bound of a two-fold dilution, we draw $h_{\text{baseline},i,s}(0)$ for each realization of the model according to

$$h_{\text{baseline},i,s}(0) \sim U(h_{\text{obs},i,s}^{\min}, 2h_{\text{obs},i,s}^{\min}). \quad (23)$$

The values of the initial latent titer, $h_{i,s}(0)$, and the initial susceptibility, $q_{i,s}(0)$, depend on the time of most recent infection, which may have occurred before entry in the study. To initialize the latent states for each individual, we draw the time of the most recent infection from the density of subtype-specific flu intensity, $L_s(t)$, in the seven years before the first observation. In this way, we account for known epidemic activity in Hong Kong before the beginning of the study (Supplementary Fig. 5). For children less than 7 y old, the distribution is truncated at birth, and the density includes the probability that the child is naive to influenza infection. For sensitivity analysis, we fitted the models using other assumptions about the density

from which we drew the time of most recent infection (see Supplementary Discussion, Supplementary Figs. 17 and 18).

**Likelihood-based inference.** The titer and infection dynamics of each individual are modeled as a partially observed Markov process (POMP). The model for each subtype is a panel POMP object, or a collection of the individual POMPs with shared age- and subtype-specific parameters. We use panel iterated filtering (PIF) to fit the models[56,57].

Iterated filtering uses sequential Monte Carlo (SMC) to estimate the likelihood of an observed time series. In SMC, a population of particles is drawn from the parameters of a given model to generate Monte Carlo samples of the latent dynamic variables. To evaluate the likelihood of a parameter set, SMC is carried out over the time series for each individual, generating a log likelihood for the corresponding panel unit. The log likelihood of the panel POMP object is the sum of the individuals' log likelihoods.

Within each PIF iteration, filtering, or weighted particle re-sampling, occurs once for all observations from each individual. One PIF iteration is one pass of the weighted re-sampling over all individuals. Damped perturbations to the parameters occur between iterations. As the amplitude of the perturbations decreases, the algorithm converges to the maximum likelihood estimate[56].

For each model, we initialize the iterated filtering with 100 random parameter combinations. We perform series of successive 50-iteration MIF searches, with the output of each search serving as the initial conditions for the subsequent search. We use 10,000 particles for each MIF search. The likelihood of the output for each search is calculated by averaging the likelihood from ten passes through the particle filter, each using 20,000 particles. We repeat the routine until additional operations fail to arrive at a higher maximum likelihood.

For model selection in the sub-model of the acute titer boost, we used the corrected Akaike Information Criterion (AICc) (Supplementary Table 1)[58]. We obtained maximum likelihood estimates for each parameter and associated 95% confidence intervals by constructing likelihood profiles. We used Monte Carlo Adjusted Profile methods[57] to obtain a smoothed estimate of the profile (Supplementary Discussion).

**Calculating imprinting probabilities.** We calculate the probability that an individual's first influenza A infection was with a particular subtype (H1N1, H3N2, or H2N2) or that the individual was naive to infection at each year of observation. We assume that the first infection occurred between the ages of 6 months and 12 years, as infants are protected by maternal antibodies for the first six months of life[59,60]

Following the original imprinting model by Gostic and colleagues[8], we estimate the probability that an individual with birth year $i$ has his or her first influenza A infection in calendar year $j$:

$$\nu_{i,j} = \frac{(1-A)^{j-1}A}{\Sigma_{j=1}^{i+12}(1-A)^{j-1}A}. \quad (24)$$

Here, $A$ is the constant annual attack rate in seronegative children as estimated by Gostic and colleagues ($A = 0.28$ [8]). Given observation year $y$, the probability that individual $i$ was first infected in year $j$ is

$$\nu_{ij|y} = \begin{cases} \frac{A}{N_{i|y}} & y \geq i+12 \\ \frac{A(\Pi_{k=1}^{j-1}(1-A))}{N_{i|y}} & y < i+12 \end{cases} \quad (25)$$

where $N_{i|y}$ is a normalizing factor that enforces the assumption that all individuals have their first infection by age 12 and ensures that all probabilities sum to one for individuals that are $\geq 12$ years old at the observation date. The normalization factor does not apply to individuals <12 years old, who have some probability of being naive to infection. We combine the probabilities of the age of first infection with annual historical influenza A subtype frequency data (see Supplementary Discussion) to determine the probability that an individual with birth year $i$ had his or her first exposure to subtype $S$ in year $j$,

$$p_{\text{imp}_{S,i|y}} = \Sigma_{j=i}^{y} f_{S|j} \nu_{i,j|y}. \quad (26)$$

Here, $f_{S|j}$ gives the fraction of specimens of subtype $S$ out of all specimens from community surveillance that are positive for influenza A. For individuals younger than 12 years old during the year of observation, the probability that an individual was naive in observation year $y$ is

$$p_{\text{naive}_{i|y}} = 1 - \Sigma_{j=i}^{y} \nu_{i,j|y}. \quad (27)$$

## Data availability

The data for this study are available at https://github.com/cobeylab/Influenza-immune-dynamics/tree/master/Data.

## Code availability

All of the software to run the analysis and produce the figures is available at https://github.com/cobeylab/Influenza-immune-dynamics. All code was built and run using R version 3.3.2.

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

## Acknowledgements

We are grateful to Phil Arevalo and Ed Baskerville for helpful discussions. We thank the University of Chicago Resource Computing Center for access to high performance computing. This project has been funded in whole or in part with federal funds from the National Institute of Allergy and Infectious Diseases (NIAID), National Institutes of Health (NIH), Department of Health and Human Services, under grant DP2AI117921 (to SC) and CEIRS Contract No. HHSN272201400005C (to SC). The household cohort study was supported by the Research Fund for the Control of Infectious Diseases of the Health, Welfare and Food Bureau of the Hong Kong SAR Government (grant numbers CHP-CE-03, 11100882 and 13120602), the Area of Excellence Scheme of the Hong Kong University Grants Committee (grant number AoE/M-12/06), and the Hong Kong Research Grants Council (grant no. T11–705/14N). Funding was also provided by NIH F30AI124636 to S.R., NIH T32GM007281, and NIGMS grant no. U54GM088558. The funders had no role in study design, data collection and analysis, decision to publish, or preparation of the manuscript.

## Author contributions

S.R., R.S., B.J.C., and S.C. designed the research; S.R. and R.S. performed the analysis; V.J. F., G.M.L., D.K.M.I, R.A.P.M.P., J.S.M.P., and B.J.C. provided data and epidemiological expertise; S.R., B.J.C., and S.C. wrote the manuscript. All authors contributed to critical review of the manuscript.

## Additional information

**Competing interests:** B.J.C. has received research funding from Sanofi, and honoraria from Sanofi and Roche. The remaining authors declare no competing interests.

