## [Peer Review File · Nature Communications]

Reviewers' comments:

Reviewer #1 (Remarks to the Author):

Referee report of the manuscript Age-specific differences in the dynamics of protective immunity to influenza by Sylvia Ranjeva and colleagues, submitted to Nature Communications.

Summary

In this manuscript, Ranjeva and colleagues fit models to longitudinal antibody responses against influenza A of 114 children and 592 adults in Hong Kong that have been followed from 2009 to 2014. The antibody responses (HI titers) provide information on the dynamics of the antibody response after infection, and on the probability of infection for a given pre-existing antibody level. The authors show that in children a model that explains susceptibility by the pre-existing HI titer is better able to describe the data than a model based on time since previous infection, while in adults a model based on time since last infection outperforms the model explaining susceptibility by the pre-existing HI titer to the currently circulating strain.

Evaluation

Initially, I had high hopes for this manuscript, which deals with the important question how the probability of infection is moulded by the pre-existing antibody response, and which draws information from a rich longitudinal data set. In principle, such data should be able to better than to date study the dynamics of HI antibody responses and probabilities of infection. However, when I finally understood what was going on (I believe the authors could have done a better job explaining the analyses) I was a bit disappointed. In essence, the authors show that in children the HI titer against the currently circulating strain is the better predictor for susceptibility, while in adults it may be the time since last infection. Given that the two predictors (HI titer against the circulating strain and time since last infection) are not independent but probably (highly) correlated, and that it is known for some time now that antibody responses in adults are more broad-reactive and geared to earlier infections (see e.g., recent papers by Florian Krammer) this is unsurprising (as HI mainly measures responses to the highly variable 4-5 dominant Ab epitopes on the head of HA and hardly any of the more conserved stalk-based epitopes). It would have been surprising if, for instance, a model with random susceptibility would have provided a better fit than either the HI or time since infection based models. In such a case, one could have made the argument that HI in adults is of little value.

Overall, therefore, it is somewhat unclear to me what is exactly new in the manuscript that would warrant publication in Nature Communications. In any case, if there are seminal novelties, I believe the authors should have stated and explained this much clearer.

Specific comments

-Figure 1 is meant to introduce and intuitively explain the main analyses and procedures, but to me it did exactly the opposite, and I was only able to understand what was going on after repeated rereading the results and methods sections. My suggestion would be to include fewer details in this figure, focussing on the main steps in the analysis.

-These data have been presented elsewhere but I believe that it would have been good if one multi-panelled figure giving an overview of the data would have been included.

-I might be mistaken but as far as I could see there were no figures included with individual fits of Ab titers over time, which would be really helpful gaining a better idea of how well the model fits to the data.

-The 706 persons have been selected from a smaller number of households and, hence, persons within households cannot be assumed completely independent. How have the authors dealt with this problem?

-p2 Table 2: Please add units.

-p4 "The models reproduce ... and other estimates of protection". strange heading. why the "other"?

-p6, Eq (5). This is a very specific choice for two specific parametric functions. How sensitive are the results to these choices?

-p6 last paragraph. "However ... tend to overestimate ... This suggests that the model might not be fully capturing ...". I did not understand this line of reasoning. To me this suggest that there is some mode misspecification. Please clarify.

-p7, first paragraph of the discussion. I agree that this may explain the main finding that in children HI is the better predictor while in adults it is the time since infection. However, with the present study the authors cannot give a definitive answer. I know that this would entail a lot of extra work (if it would be feasible at all) but it would be fascinating if direct measrements would be available of the strength of the response to conserved HA stalk-based antibodies (e.g., using micro array based tests).

-p7 last sentence of the second paragraph. Strange sentence ("if there are indeed persons who never get the ful"). Either drop or explain in more detail.

-p11 Monte Carlo adjusted profile confidence intervals. I was a bit alarmed by the sometimes large uncertainties in the limits of the confidence intervals, e.g., in Figures S3b, S4a, S5, S18, and S20. This is also true for the ML values. How does this affect the main results?

Recommendation

Based on the above evaluation that acknowledges that the research question is important and the available data unique, but also points to shortcomings in the analyses and reasoning, I recommend either a major revision, or a rejection with encouragement to resubmit if the authors would be able to address the major points raised above.

Reviewer #2 (Remarks to the Author):

The authors use data from a longitudinal serosurvey to compare how well different assumptions about susceptibility can reproduce observed patterns. The model suggests that risk of infection for children is mediated by current HI titre, but for adults, a non-HI-specific value (time since infection) is a better predictor of risk. The authors also find no evidence of HA group imprinting on infection risk.

I found this to be a thorough and thoughtful study, making good use of a nice dataset. However, I did find some aspects of the analysis hard to follow given the (albeit necessary) complexity of the model. I also had some queries about the fitting process, and robustness of the estimates.

Detailed comments:

- Was the data on PCR-confirmed infections used in fitting the full model as well as the sub-model? I couldn't see this mentioned in the text. If the PCR confirmations were not used in the fitting, it could provide a validation of the inference of infection times.

- I found the infection part of the model hard to piece together. For example, how does the temporal risk of infection (λ) in Equation 6 come into the likelihood? What is the relationship between the $t_{\{i,s\}}$ terms and λ ?

- If I understand correctly, the model estimates latent infection times, and there is also some data on symptom reporting for participants. Is it therefore possible to obtain an estimate of how many infections are asymptomatic? The values in Table S6 are difficult to generalise given that they depend on the study design, but it would be interesting if the model could provide an estimate of how many infections are likely to go undetected, and how serological dynamics relates to symptom reporting.

- In Fig S3, there is quite a lot of variation in the likelihood as the parameters vary. In some cases for H3N2, there is a difference of almost 10 log-likelihood points for similar parameter values. Is this an issue with convergence of the MIF algorithm? Or something to do with the structure of the data/model? Such differences are potentially concerning, given that some of the models compared in Table 1 only have 20–25 a difference in AICc.

- How well did the model fit the individual-level titres? Figs S7–S8 compare the distributions of rises, but it would be good to see a histogram of observed minus estimated titres at time of sampling across all participants.

- The combined susceptibility function in Equation 3 seemed a bit inelegant, given that it can produce an abrupt switch in the dominant component of protection (as seen in Fig 2). Did the authors consider a function with a smoother transition, like a weighted average of the two?

- What was the logic for normalizing titre boosts by the gap between samples in Fig S15?

Minor comment:

- It took me a while to work out precisely what the authors meant by infection, susceptibility and protection in the model. As I understand it, they define infection as an exposure to influenza that generates a detectable antibody response, susceptibility is the probability an exposure will result in such an infection, and non-susceptibility is equivalent to protection. However, it would be helpful to explicitly define these terms (e.g. protection could be interpreted as protection from disease rather than serological response).

Reviewer #3 (Remarks to the Author):

This manuscript describes the dynamics of protective immunity to influenza virus infection in a household cohort study. The main finding is that protective immunity in children is strongly associated with antibody titer (HI) levels, whereas the protective immunity in adults is weakly associated with antibody titer levels. The data set is unique and allows for a much deeper understanding of the immunity dynamics than before. The statistical modelling of the data is complex, which is inherent to the complexity of the system being modelled here.

Overall, this manuscript presents an impressive effort towards a better understanding of protective immunity against influenza virus infection. However, there is an important issue that needs to be addressed.

It is not clear that the age-specific exposure to influenza virus is modelled correctly. An individuals' hazard rate of exposure is modelled as the product of an age-specific susceptibility per infectious contact event, the age-specific rate of contact events, times the probability that the contact is made with someone who is infectious. (in the manuscript the term risk is used, but I believe this must be hazard rate). The probability that contact is made with someone who is infectious is proportional to the proportion of ILI consultations among all GP consultations. The age-specific contact rates are based on observed social contact data (reference 49). The model ignores the age

of the infectors. This introduces an inconsistency: it is supposed that the incidence of influenza virus infection of infectees varies with age, whereas the incidence of influenza virus infection of infectors does not vary with age. What I would have expected is the use of the observed contact rate matrix and a vector that reflects the typical distribution of infections over age times a time-varying function that describes the intensity of influenza circulation. Without this, the model includes an inherent inconsistency in the age-specific exposure to influenza virus. The key argument of the manuscript requires the age-specific exposure to be correct. Please inspect the model, correct it where needed, and discuss alternative explanations of the findings.

Minor issues:

It is not clear from the text how susceptibility should be interpreted. From the methods section I understand it is a variable on a scale from 0 (complete immunity) to 1 (complete susceptibility). But what does a susceptibility of 0.3 mean? An individual with a susceptibility of 0.3 could have a probability of 0.3 of being completely immune, or this individual could have a probability of 0.3 being infected when exposed during each exposure?

The link is missing between the dynamics of the latent titer $h_{i,s}(t)$ and the likelihood function which is a function of $\log H_{i,s}$. Is the missing equation $\log H_{i,s} = \log_2 (h_{i,s} / 10) + 2$, or $\log H_{i,s} = \log(h_{i,s})$?

The text could be made more precise. For example: "epidemic incidences range from 5-17% in adults" requires a statement about the time unit for measuring incidence. The text and the equations do not always match. For example as "an exponential function that starts at 0" and "the individual trajectories in fig 2 (A,B) are constrained from above by the exponential curve" but equation 8 that the function is $1 - \exp()$ rather than $\exp()$.

Reply to reviewers

We begin by describing the changes to the model and present an overview of the revised results. We then provide point-by-point responses to the reviewers' specific comments. Revised text in the manuscript and quotes from the manuscript in the reply appear in blue.

Structural changes to the model

The referees suggested three structural changes, which we incorporated:

1. To better account for age-specific rates of exposure to infection, we modified the community-level infection rate by the distribution of infections among age-specific contacts.
2. We included household structure. We allow the instantaneous per capita rate of infection to increase with the presence of infected household members.
3. To simplify the inference, rather than comparing the performances of models with HI-correlated vs. non-HI-correlated protection, we now model the instantaneous susceptibility of infection, $q_{i,s}(t)$ as the weighted average of HI-correlated and non-HI-correlated protection. We had previously modeled protection as the minimum of the two components.

Revised model of the instantaneous per capita rate of infection

All of the reviewers' suggested changes affect the per capita hazard rate of infection. The instantaneous per capita infection rate for individual i with subtype s , $\lambda_{i,s}(t)$, comprises a rate of infection from the community, $\lambda_{\text{community},i,s}$, and another from the household, $\lambda_{\text{household},i,s}$,

$$\lambda_{i,s}(t) = \lambda_{\text{community},i,s}(t) + \lambda_{\text{household},i,s}(t). \quad (1)$$

We now incorporate the age distribution of infectious contacts, such that the force of infection for individual i with subtype s resulting from contact with individuals in the community, $\lambda_{i,s,\text{community}}(t)$, is

$$\lambda_{\text{community},i,s}(t) = q_{i,s}(t) \beta_{\text{scaled},s} L_s(t) \sum_{n=1}^{N_{\text{cat}}} \beta_{\text{c},\text{cat}_i,\text{cat}_n} p_{\text{cat}_n}. \quad (2)$$

Here, $q_{i,s}(t)$ is individual i 's susceptibility to subtype s at time t , $\beta_{\text{c},\text{cat}_i,\text{cat}_n}$ is the fixed contact rate for an individual of age category cat_i with individuals of age category cat_n (Leung *et al.* 2017), $L_s(t)$ the subtype-specific community-level flu intensity, N_{cat} is the number of contact age categories, and p_{cat_n} is the fraction of influenza infections attributable to age class cat_n (Caini *et al.* 2018 *BMC Inf. Dis.*).

Individual i also experiences a within-household infection rate

$$\lambda_{\text{household},i,s}(t) = q_{i,s}(t) \omega_s I_{\text{other},s}(t) \quad (3)$$

where $q_{i,s}(t)$ is the individual's susceptibility and $I_{\text{other},s}(t)$ is an indicator variable that equals 1 if any other household member is infected with subtype s at time t and 0 if not. Because we do not track households in their entirety (we sample only individuals that were not vaccinated during the study), we do not model density-dependent within-household transmission. Rather, ω_s describes the daily influenza exposure rate that individual i experiences with the presence of any infected family member.

Under the weighted susceptibility model, individual i 's subtype- s -specific susceptibility at time t is the weighted average of the susceptibility determined by HI-correlated and non-HI-correlated protection,

$$q_{i,s}(t) = q_{\text{HI-correlated},i,s}(t) \psi_{a_i,s} + q_{\text{non-HI-correlated},i,s}(t) (1 - \psi_{a_i,s}), \quad (4)$$

where $\psi_{a_i,s}$ weights the contribution of HI-correlated protection to susceptibility in children ($a_i = \text{children}$) and adults ($a_i = \text{adults}$). The value of $\psi_{a_i,s}$ gives us the best-fit form of protection, with $\psi_{a_i,s} = 0$ and $\psi_{a_i,s} = 1$ corresponding to completely non-HI-correlated and completely HI-correlated protection, respectively.

Subtype	Parameter		MLE [95% CI]	Previous MLE [95% CI]
H3N2	Long-term boost	$\zeta_{adults,s}$	0.0 [0.0, 0.03]	0 [0, 0.01]
		$\zeta_{children,s}$	0.04 [0.02, 0.07]	0.02 [0, 0.04]
	Weight of HI-correlated immunity	$\psi_{adults,s}$	0.0 [0.0, 0.2]	N/A
		$\psi_{children,s}$	1.0 [0.8, 1.0]	N/A
	50% protective titer	$TP50_{children,s}$	44 [29, 74]	60 [42, 122]
	Half-life non-HI-correlated immunity	from $w_{nonspecific,adults,s}$	4.1 y [3.2, 5.5]	2.1 y [1.3, 3.3]
Within-household transmission rate	ω_s	0.05 [0.02, 0.11]	N/A	
H1N1pdm09	Long-term boost	$\zeta_{adults,s}$	0.06 [0.05, 0.07]	0.01 [0, 0.03]
		$\zeta_{children,s}$	0.3 [0.2, 0.5]	0.2 [0.1, 0.4]
	Weight of HI-correlated immunity	$\psi_{adults,s}$	0.10 [0.07, 0.12]	N/A
		$\psi_{children,s}$	1.0 [0.8, 1.0]	N/A
	50% protective titer	$TP50_{adults,s}$	8 [1, 12]	540 [90, 5120]
		$TP50_{children,s}$	15 [8, 25]	30 [10, 45]
Half-life non-HI-correlated immunity	from $w_{nonspecific,adults,s}$	4.0 y [3.1, 5.2]	3.4 y [2.6, 4.7]	
Within-household transmission rate	ω_s	0.06 [0.03, 0.09]	N/A	

Table 1: Maximum likelihood estimates and 95% confidence intervals (CI).

Summary of new results

Our main conclusions are unchanged by the changes in model structure. We find that, for both H1N1pdm09 and H3N2, the dynamics of protection are best explained by HI-correlated protection in children, while non-HI-correlated protection dominates in adults. We identify a minor role for HI-correlated protection in adults for H1N1pdm09, which is absent for H3N2.

Weight of HI-correlated protection and duration of protection after infection

In children, the model favors completely HI-correlated protection for both H1N1pdm09 and H3N2 (MLE for $\psi_{children,s} = 1$, Table 1). The estimated 50% protective titers in children are slightly lower than our previous estimates for H1N1pdm09 ($TP50_{children,s} = 15$ [8, 25]) and for H3N2 ($TP50_{children,s} = 44$ [29, 74], Table 1).

To estimate the duration of protection in children with each subtype, we simulated the latent post-infection susceptibility from the model at the best-fit parameters (Fig. 1). We find again that because susceptibility in children depends on antibody titers, the duration of protection in children varies with their pre-infection titers and differences in the magnitude of the acute boost post-infection. The mean effective half-lives of protection in children are 3.5 y (95% quantile: [1.4, 5.2]) against H3N2 and 7.1 y (95% quantile: [2.8, 8.8]) against H1N1pdm09 (Fig. 1 C,D), but there is substantial variation between individuals.

In adults, the model favors completely non-HI-correlated protection for H3N2 ($\psi_{children,s} = 0.0$ [0.0, 0.2]). HI contributes slightly to protection in adults for H1N1pdm09 ($\psi_{adults,s} = 0.1$ [0.07, 0.12]). The low 50% protective titer in adults ($TP50_{adults,s}$) for pH1N1 (8 [1, 12]) implies that most individuals (i.e., even individuals with low titer) benefit from some additional protection after infection. From the simulations of the latent susceptibility after infection, it is clear that, consistent with our previous results, susceptibility to both H3N2 and H1N1pdm09 in adults is driven primarily by the time since infection rather than by the titer (Fig. 1). Protection against H3N2 in adults wanes according to the rate of non-HI-correlated protection, with a half-life of 4.1 y. The contribution of HI titer to post-infection protection to H1N1pdm09 creates mild individual-level variability according to pre-infection titers. In aggregate, adults' susceptibility to H1N1pdm09 thus wanes with a half-life of approximately 6.4 y (95% quantile: [4.8, 6.7]), reflecting a slower loss of protection than would be predicted under purely non-HI-correlated protection (Table 1).

In both age groups, protection wanes faster against influenza A(H3N2) than A(H1N1)pdm09. Protection against H3N2 lasts longer in adults than in children.

Simulated epidemic activity over the study period

The models again recover reasonable population-level patterns of infection for both subtypes (details in the main text, "Discussion"). The inferred monthly subtype incidences from the simulated latent infections (Fig. 2, Table 2) are largely unchanged. The new estimates remain comparable to the monthly incidence calculated from the subtype-level population intensity (Fig. 2) and consistent with other estimates of epidemic activity (Sullivan *et al.* 1993 *Am J Pub Health*, Tokars *et al.* 2018 *Clin Inf Dis*).

Figure 1: Susceptibility after simulated infection (at time $t = 0$) for adults with H1N1pdm09 (a) and H3N2 (b) and for children with H1N1pdm09 (c) and H3N2 (d). The black lines represent individual trajectories over the study period from five simulations. The red line represents the median among individuals over 1000 replicate simulations. Curves from individual trajectories are truncated at points corresponding to the end of the study.

Figure 2: Simulated monthly incidence of H1N1pdm09 (a, bottom) and H3N2 (b, bottom) in children and adults, averaged over 1000 simulations, contrasted with respective monthly community intensities (a and b, top). The shaded areas are bounded by the 2.5% and 97.5% quantiles from the simulations. Horizontal black bars denote inter-epidemic periods for odds ratios (OR), indexed by epidemic.

Subtype		Epidemic	Simulated incidence	From observed ≥ 4 -fold changes
H1N1pdm09	Adults	1	0.09 [0.06, 0.16]	0.05
		2	0.12 [0.09, 0.15]	0.12
		3	0.07 [0.04, 0.09]	0.06
	Children	1	0.14 [0.09, 0.33]	0.10
		2	0.17 [0.12, 0.24]	0.19
		3	0.10 [0.05, 0.15]	0.06
H3N2	Adults	1	0.12 [0.09, 0.15]	0.16
		2	0.13 [0.11, 0.16]	0.15
		3	0.04 [0.03, 0.05]	0.03
	Children	1	0.22 [0.15, 0.30]	0.21
		2	0.23 [0.16, 0.30]	0.24
		3	0.06 [0.02, 0.13]	0.03

Table 2: Incidence in each epidemic (Fig. 2). The simulated incidence was estimated from the latent simulated infections. The main and bracketed values give the median and 95% quantiles, respectively, from 1000 replicate simulations of the models at the maximum likelihood estimate. Incidence was also estimated from the frequency of ≥ 4 -fold titer consecutive titer rises observed in the data.

Referee 1

Evaluation

Initially, I had high hopes for this manuscript, which deals with the important question how the probability of infection is moulded by the pre-existing antibody response, and which draws information from a rich longitudinal data set. In principle, such data should be able to better than to date study the dynamics of HI antibody responses and probabilities of infection. However, when I finally understood what was going on (I believe the authors could have done a better job explaining the analyses) I was a bit disappointed. In essence, the authors show that in children the HI titer against the currently circulating strain is the better predictor for susceptibility, while in adults it may be the time since last infection. Given that the two predictors (HI titer against the circulating strain and time since last infection) are not independent but probably (highly) correlated, and that it is known for some time now that antibody responses in adults are more broadly reactive and geared to earlier infections (see e.g., recent papers by Florian Krammer) this is unsurprising (as HI mainly measures responses to the highly variable 4-5 dominant Ab epitopes on the head of HA and hardly any of the more conserved stalk-based epitopes). It would have been surprising if, for instance, a model with random susceptibility would have provided a better fit than either the HI or time since infection based models. In such a case, one could have made the argument that HI in adults is of little value. Overall, therefore, it is somewhat unclear to me what is exactly new in the manuscript that would warrant publication in Nature Communications. In any case, if there are seminal novelties, I believe the authors should have stated and explained this much clearer.

We are encouraged by Referee 1's enthusiasm for the research question. We respect Referee 1's point that the novelty of the findings could be clarified and have revised the manuscript in response. The changes are summarized here. In a nutshell, the main novel contributions of this work are to provide epidemiological evidence that different forms of immunity protect children and adults and to estimate the durations of protection to H1N1pdm09 and H3N2. The study also shows that immune "imprinting" (Gostic et al., 2016) does not protect much against non-medically attended infections; estimates negligible long-term cross-immunity between subtypes; and quantifies relationships between individuals' antibody titer dynamics, protective immunity, and epidemic patterns.

As Referee 1 notes, an emerging hypothesis in influenza immunology is that antibody responses change with age to target conserved epitopes, such as epitopes on the HA stalk. Serum titers of broadly neutralizing HA stalk-reactive antibodies increase with age (Miller et al. 2013 *Sci. Trans. Med.*). Nachbagauer, Krammer, and colleagues (Nachbagauer et al. 2018 *Nat. Immunol.*) found that infection with pandemic H1N1 induced a broader antibody response than infection with seasonal H3N2 in a small hospital cohort, suggesting that secondary exposure to divergent HA head proteins of a previously encountered subtype induces antibodies to conserved epitopes. In the same study, using cross-sectional sera from a separate cohort, the authors found that younger people exhibited strong, narrow responses against recent H3N2 and H1N1 strains, whereas middle-aged and older individuals exhibited more cross-reactive responses directed towards strains encountered in childhood. This result suggests that the cross-reactivity of the antibody response increases with age and repeated infections in a way that is influenced by early exposures. Andrews et al. (2015 *Sci. Trans. Med.*) reported

stalk-directed responses after initial vaccination with H1N1pdm09. However, these responses became more head-reactive with successive vaccinations.

These observations describe the dynamics of immune memory, but the consequences for protection have been largely unexamined. This work provides evidence that protection in children is largely mediated by immunogenic epitopes near the HA head and that protection in adults is not. Although perhaps unsurprising, this has not to our knowledge been shown before. Furthermore, our estimation of the duration of subtype-specific protection after infection provides evidence that despite differences in antibody specificity, there are no noticeable differences in the duration of this protection in adults and children. Furthermore, these estimates themselves are important. To our knowledge, they have been estimated few times before (Kucharski *et al.* 2015 and 2018 *PLOS Biol.*), and using indirect approaches.

Weak to absent imprinting is interesting because this is one of the few data sets available where the rate of medically unattended infections can be estimated. Thus, the model implies that imprinting affects disease severity more than susceptibility to infection.

Understanding the strength and duration of competition between H3N2 and H1N1 is important for epidemic forecasting and developing theories of influenza diversity. Previously, competition has been inferred indirectly from immunological studies, population models, and brief (single-season) longitudinal epidemiological studies. The latter two have focused on temperate populations. Our model shows that cross-subtype protection, if it exists, fades quickly.

This work makes a new link between individual-level antibody responses and the dynamics of protective immunity. Previous population-level transmission models that couple influenza-like-illness time series to antigenic changes in seasonal viruses (e.g. Du *et al.* 2017 *Sci. Trans. Med.*) demonstrate the importance of waning protection due to antigenic evolution for inter-annual epidemic dynamics. These models estimate rates of immune waning consistent with phylogenetic and epidemic trends. However, population-level transmission models have not explicitly estimated the duration of protection after influenza infection at an individual level or explored individuals' antibody responses to infection. Conversely, previous models of HI titer dynamics have quantified individuals' antibody responses to influenza infection at multiple timescales (e.g. Kucharski *et al.* 2015 *PLOS Biol.*, Lessler *et al.* 2012 *PLOS Path.*, Kucharski *et al.* 2018 *PLOS Biol.*) but have not separated the antibody dynamics from the dynamics of immune protection. Here, we estimate the duration of immune protection at the individual level separately from the antibody dynamics. Simulations of the individual-level immune processes from our model then give rise to the population epidemic dynamics, including realistic attack rates.

A technical comment on identifiability: Referee 1 suggests the results are partly unsurprising because HI titers correlate so well with time since infection. In the original model, in adults, models containing only HI-correlated protection performed dramatically worse ($\Delta AICc > 40$) for both subtypes than models that excluded HI-correlated protection. This result would not have been possible the two forms of protection were highly correlated, since high correlation would lead to non-identifiability. (In the current model, the analogue would be a wide confidence interval for the parameter that weights each form of immunity.) Additionally, infection in the model is driven by an external proxy of community flu intensity, limiting confounding in the infection rate from the HI titers themselves. If the dynamics were entirely driven by the epidemic intensity, and neither the titer nor the time since infection affected susceptibility, then the parameters that modify susceptibility and the infection rate based on the individual-level immune dynamics would similarly be non-identifiable. Instead, we find they are informative.

Specific comments

1. Figure 1 is meant to introduce and intuitively explain the main analyses and procedures, but to me it did exactly the opposite, and I was only able to understand what was going on after repeated rereading the results and methods sections. My suggestion would be to include fewer details in this figure, focusing on the main steps in the analysis.

We have tried to make main text Fig. 1 clearer (here, Fig. 3).

Figure 3: Schematic of modeling approach. Steps 1 and 2 are performed for each subtype. **Step 1:** (a) First, the sub-model of short-term post-infection titer dynamics is fitted to a subset of the data. This subset includes the time of PCR-confirmed infection and the immediate pre- and post-infection titers. The mean and standard deviation of the acute titer boost and its dependence on pre-infection titer (the antibody ceiling effect) are fitted. (b) Next, fixing the parameters associated with short-term titer dynamics (Step 1a), the full model is fitted to titers from the entire cohort. The contribution of HI-correlated and non-HI-correlated protection, the titer waning rate, the 50% protective titer, and the long-term boost after infection are estimated. **Step 2:** (a) The duration of protection and inter-epidemic protection are estimated from simulating population-level dynamics from the best-fit model in Step 1b. From the latent infections and susceptibility for each individual, we track the loss of protection after infection. We also estimate the incidence and the odds ratios (OR) of protection between epidemics. (b) Simulation enables additional checks of the model. We compare the simulated and observed distributions of n -fold titer rises and coefficients of titer variation among individuals.

Reviewer 1 also commented that “the authors could have done a better job explaining the analyses.” We have made many small changes, including to the introduction to the model:

(lines 60 - 64) To estimate the contribution of HA-head-directed antibodies to protection from influenza infection in children and adults, Eq. 4 was incorporated into a partially observed Markov model that simulates individuals’ latent (unobserved) HI titers and susceptibility to infection over time while simultaneously accounting for measurement error (Fig. 1, Step 1b; Methods). The model assumes that infection can change the antibody titer, which allows infection events and thereby latent susceptibility ($q_{i,s}(t)$) to be inferred from longitudinal sera.

We also clarified the role of the sub-model:

(lines 66 - 68) In the model, infection acutely boosts an individual’s titer, which then wanes slowly over one year, potentially leaving a long-term boost that does not wane. To increase accuracy in modeling these acute boosts, we took advantage of 112 PCR-confirmed infections and pre- and post-infection titers from this study to fit the mean and standard deviation of the titer rises (Fig. 3, Step 1a; section S1) ... (lines 71 - 75) After fitting this “sub-model” to describe the relationship between infection and short-term titer changes, we then fixed its parameters to fit the full model of titer dynamics to all 706 individuals... The full model estimates the contribution of HI-correlated and non-HI-correlated factors to protection (Eq. 3), the magnitude of the long-term titer boost, the 50% protective titer (for Eqs. 1 and 3), and the rate of waning of non-HI-correlated protection (for Eqs. 2 and 3) (Fig. 1, Step 1b).

We have also clarified details in the “Complete model description” (lines 237 - 315; discussed in our replies to Referee 2, point 2; Referee 2, minor comment, point 1; and Referee 3, point 2).

2. These data have been presented elsewhere but I believe that it would have been good if one multi-paneled figure giving an overview of the data would have been Included.

We now include a figure summarizing the data: it shows the community-level flu intensity during the study, the household structure, and the numbers of titers and PCR-confirmed infections over time (here, Fig. 4).

Figure 4: Features of the raw data. (a) Community intensity of H3N2 and H1N1pdm09 (ILI x %positive) over the study (b) Distribution of household sizes among the study cohort (c) Number of titer samples for H3N2 and H1N1pdm09 in children and adults over the study. (d) Number of PCR-confirmed infections with H3N2 and H1N1pdm09 in children and adults over the study.

3. I might be mistaken but as far as I could see there were no figures included with individual fits of Ab titers over time, which would be really helpful gaining a better idea of how well the model fits to the data.

We thank Referee 1 for highlighting an important point of clarification. We use simulation-based maximum likelihood inference for longitudinal partially observed Markov processes. In our model, the composite dataset is the collection of individuals' titers, where the individual time series are dynamically independent apart from the shared model parameters. These shared parameters, such as the age- and subtype-specific rates of immune waning $w_{\text{nonspecific,adults},s}$ and the subtype-level transmission rate $\beta_{\text{scaled},s}$, guide the dynamics (e.g. the epidemic behavior and the distribution of titer changes) in aggregate. Therefore, simulations from the model reproduce summary features of the dataset, including epidemic trends (Fig. 2) and the distribution of titer changes among individuals (Figs. 5, 6). "Nuisance" parameters, such as the inferred infection times for different individuals, are not implicitly generated by our method the way they are under Bayesian approaches. Nonetheless, we show here the filtered particle population at the MLE for a few individuals (Figs. 7, 8), which is the closest analogue to the Bayesian posterior, although the MLE obviously does not capture the full uncertainty in parameter estimation.

Figure 5: Observed and simulated distributions of consecutive 2-, 4-, and 8-fold titer rises per individual in the H1N1pdm09 data. The dashed blue lines give the medians from 1000 replicate simulations of the model, and the shaded blue areas are bounded by the 2.5% and 97.5% quantiles.

Figure 6: Observed and simulated distributions of consecutive 2-, 4-, and 8-fold titer rises per individual in the H3N2 data. The dashed blue lines give the medians from 1000 replicate simulations of the model, and the shaded blue areas are bounded by the 2.5% and 97.5% quantiles.

Figure 7: Simulated individual trajectories from the filtered particle population of the model for H1N1pdm09 at the maximum likelihood parameters. The solid and dashed black lines give the observed log titer and the filtered log titer trajectory, respectively. The dashed red lines denote latent times of infection from the model. Results are shown for the first six individuals in the dataset.

Figure 8: Simulated individual trajectories from the filtered particle population of the model for H3N2 at the maximum likelihood parameters. The solid and dashed black lines give the observed log titer and the filtered log titer trajectory, respectively. The dashed red lines denote latent times of infection from the model. Results are shown for the first six individuals in the dataset.

4. The 706 persons have been selected from a smaller number of households and, hence, persons within households cannot be assumed completely independent. How have the authors dealt with this problem?

We now include households. For both H1N1pdm09 and H3N2, the revised model (Eq. 3) shows a within-household risk effect (the 95% CIs for the within-household effect ω_s exclude zero, Table 1, Fig. 9). We state in the main text:

(lines 123 - 128) We estimate that for both subtypes, the daily within-household intensity (the increased risk of transmission given that at least one member in the household is infected) is roughly 5%, or $\sim 25\%$ over an average five-day infectious period (ω_s , Table 1). This is consistent with published estimates of secondary household attack rates for influenza A virus ranging from 10% to 30% (Longini *et al.* 1982 *Am J Epidemiol.*, Viboud *et al.* 2004 *B J. Gen. Pract.*, Hayden *et al.* 2004 *J. Inf. Dis.*, Petrie *et al.* 2013 *PLoS ONE*, Savage *et al.* 2011 *BMC Pub. Health*, Papenburger *et al.* 2010 *Clin. Inf. Dis.*). For example, for a young child at the peak of a H1N1pdm09 epidemic, the daily risk of transmission from the community is approximately 2.1% per day. Therefore, our results suggest that the within-household transmission rate is at least twice as high as the maximum community-level risk. The same is true for H3N2.

Figure 9: Likelihood profiles of the household risk parameter, ω_s , for H1N1pdm09 and H3N2. The vertical dashed lines denote the 95% CIs.

5. p2 Table 2: Please add units.

Units have been added to Table 2.

6. p4 “The models reproduce . . . and other estimates of protection”. strange heading. Why the “other”?

We have removed the term “other” from the section heading.

7. p6, Eq (5). This is a very specific choice for two specific parametric functions. How sensitive are the results to these choices

In the original model, an individual’s instantaneous subtype-specific susceptibility was determined by the minimum susceptibility of HI-correlated protection and non-HI-correlated protection. We relied on model selection via the corrected Akaike Information Criteria to distinguish between models that accounted for HI-correlated protection, non-HI-correlated protection, or both. In the revised model, we use a smooth, weighted-susceptibility term that implicitly contains each of these hypotheses. This model is more flexible but yields the same results (see “Summary of new results.”). There is good outside evidence that the relationship between susceptibility to infection and HI is well represented by a logit curve (Hobson *J. Hyg.* 1972; Black *Ped. Inf. Dis. J.* 2011; Coudeville *BMC Med. Res. Meth.* 2010).

8. p6 last paragraph. “However . . . tend to overestimate . . . This suggests that the model might not be fully capturing . . .” I did not understand this line of reasoning. To me this suggests that there is some mode misspecification. Please clarify.

We are pointing out a limitation of the fitted model: it slightly overestimates variation in an individual’s titer over time. To prevent overfitting and make the inference tractable, the model does not try to fit individual-specific parameters, other than an individual’s baseline titer. All other parameters governing dynamics are shared by the entire population or the entire population of children or adults. The median number of titer samples per individual is 9. Models that estimate individual differences in titer variability will require more samples. We have flagged this future direction in the Discussion. Nonetheless, the model’s error here appears small. We compared the observed and simulated distributions of the coefficient of variation (CV) of individuals’ titers over time for children and adults with both subtypes (main text Figs. S8 and S9). We now note in the text:

(section S2) ...the difference in the means of the observed and simulated CV distributions ranges from 0.0 to only 0.1 for children and adults with H1N1pdm09 and H3N2. Furthermore, the filtered particle population of the model, an estimate of the smoothed distribution of latent model variables, at the maximum likelihood parameters closely reproduces the observed titer trajectories for individuals (Figs. 7, 8).

While we find it important to describe this observation, the model’s ability to reproduce every other pattern we have examined inspires confidence.

9. p7, first paragraph of the discussion. I agree that this may explain the main finding that in children HI is the better predictor while in adults it is the time since infection. However, with the present study the authors cannot give a definitive answer. I know that this would entail a lot of extra work (if it would be feasible at all) but it would be fascinating if direct measurements would be available of the strength of the response to conserved HA stalk-based antibodies (e.g., using micro array based tests).

We would love more assays of antibody specificity, but not enough sera is left for such analyses. Our study nonetheless provides indirect evidence that is consistent with an emerging hypothesis in immunology, and critically, one that has been previously unlinked to individual protection and epidemic dynamics. Extrapolating susceptibility from HI titers has been a cornerstone of epidemiological surveillance, e.g., for establishing correlates of protection (e.g. Hobson *et al.* 1972 *J Hygiene London*). Multiple statistical models fitted to paired pre- and post-epidemic HI titers (e.g. Zhao *et al.* 2017 *Am. J. Epidemiol.*, Psu *et al.* 2014 *BMC Inf. Dis.*) have estimated susceptibility from HI titers alone. Our model advances the field by relaxing the assumption that susceptibility correlates entirely with HI titer and by allowing for multiple, potentially unobserved infections between observations. The discovery that time since infection correlates better with protection in adults should motivate further immunological investigations of the dynamics of protection to influenza.

10. p7 last sentence of the second paragraph. Strange sentence (“if there are indeed persons who never get the flu”). Either drop or explain in more detail.

We have removed this phrase without loss of meaning.

11. Monte Carlo adjusted profile confidence intervals. I was a bit alarmed by the sometimes large uncertainties in the limits of the confidence intervals, e.g., in Figures S3b, S4a, S5, S18, and S20. This is also true for the ML values. How does this affect the main results?

Variance in the likelihood arises from the high Monte Carlo error in the inference of stochastic dynamics in large datasets. The Monte Carlo Adjusted Profile technique (Ionides *et al.* 2017 *J. R. Soc. Interface*) was developed and validated for robust parameter estimation in this setting. In the new section S3, we describe the Monte Carlo error and the MCAP approach:

A central feature of inference of stochastic models from large datasets is non-negligible Monte Carlo error that often makes it infeasible to calculate the likelihood with an error of less than one log likelihood unit. This principle holds especially for longitudinal (or panel) data, which often consist of a collection of individual time series that are dynamically independent apart from shared model parameters. Standard approaches for constructing 95% confidence intervals (CIs) rely on a threshold of 1.92 log likelihood units from the maximum log likelihood to construct the CI (Bolker 2007). Therefore, high Monte Carlo error, or error in the likelihood calculation, also poses a challenge for estimating 95% confidence intervals. The Monte Carlo Adjusted Profile (MCAP) technique (Ionides *et al.* 2017 *J R Soc Interface*) shows that a quadratic approximation in the region of the maximum likelihood can be used to reliably extrapolate the 95% CI in systems with high MC error. The MCAP algorithm accounts for the Monte Carlo error in the standard error of the spline fitted to a given likelihood profile. Importantly, Ionides and colleagues have shown that, despite wider uncertainty around the maximum likelihood in systems with high Monte Carlo error, the MCAP approach reliably identifies the parameters of the underlying dynamical system. While the Monte Carlo variance of the log-likelihood estimate increases linearly with the amount of data, so too does the Fisher information, or the information provided about the underlying dynamic system, and therefore the ability to reliably identify the maximum likelihood parameters.

Referee 2

Remarks to the Author: I found this to be a thorough and thoughtful study, making good use of a nice dataset. However, I did find some aspects of the analysis hard to follow given the (albeit necessary) complexity of the model. I also had some queries about the fitting process, and robustness of the estimates.

Specific comments

1. Was the data on PCR-confirmed infections used in fitting the full model as well as the sub-model? I couldn't see this mentioned in the text. If the PCR confirmations were not used in the fitting, it could provide a validation of the inference of infection times.

The full model was fitted only to serology. The small set of available PCR-confirmed infections and their accompanying titers (pre- and post-infection) were used to estimate the acute titer boost after infection in the sub-model. This acute titer boost was then fixed for the full model, the other components of which were fitted to all available serology. As explained in the reply to Referee 1, point 3, the frequentist approach does not infer each individual's sequence of infections (the nuisance parameters), and thus cannot predict the infection times for a particular individual.

2. I found the infection part of the model hard to piece together. For example, how does the temporal risk of infection (λ) in Equation 6 come into the likelihood? What is the relationship between the $t_{i,s}$ terms and λ ?

We thank Referee 2 for helping us improve the clarity of the paper. The infection and titer dynamics for each individual are a partially observed Markov process. Each individual's latent (unobserved) states are the infection status, the time since infection, and the true titer. The only role of the latent states is to contribute to the likelihood of observing a titer (the data): this occurs through the measurement model, which calculates the likelihood of the data at a given time point, given the underlying latent state (the true titer). In sequential Monte Carlo, these likelihoods are calculated via simulation of the latent states for distinct sets of parameters. The instantaneous per capita infection rate, $\lambda_{i,s}(t)$, is part of the underlying Markov model of the latent titer dynamics. This rate determines individual infection rate and does not contribute directly to the likelihood. However, individuals with high rates of infection may become infected, and thus may have their latent titers boosted, which could make the observed titers more or less likely. We have revised the "Complete model description" (lines 256 - 355):

The model simulates the titer and infection dynamics for each individual. Briefly, an individual's instantaneous infection rate or force of infection, $\lambda_{i,s}(t)$, is determined by the rate of exposure to infectious contacts in the community and household and by the individual's susceptibility, defined as the probability of infection per infectious contact. Infections occur stochastically and can change the latent titers. Simulating the model generates a latent titer for each individual at each observation. The measurement model then calculates the likelihood of each observed titer given the contemporaneous latent titer. The log likelihood of the model under any set of parameters is then the sum of the log likelihoods across individuals, which are the sums of the log likelihoods across observations.

The new equations 21 and 22 (lines 337 - 343) explicitly define the likelihood of the model for a given parameter set as the sum of the likelihoods across individuals.

For each individual, the log likelihood is the sum of the log likelihoods across observations

$$\mathcal{L}_i = \sum_{n=1}^{n_{\text{obs}}} \mathcal{L}(j | \theta, \log H_{i,s}(t_n)) \quad (5)$$

where n_{obs} is the number of observations for individual i and t_n gives the time at observation n . The log likelihood of the model for any parameter set θ is then the sum across n_{ind} individuals

$$\mathcal{L} = \sum_{i=1}^{n_{\text{ind}}} \mathcal{L}_i \quad (6)$$

The $t_{i,s}$ terms (Eqs. 7 - 15) are latent states that track the time of infection ($t_{i,s}^X$) and peak antibody response ($t_{i,s}^P$) in the simulations of the latent titer $h_{i,s}(t)$. They affect $\lambda_{i,s}(t)$ only insofar as they affect the titer $h_{i,s}(t)$, but they are not fitted parameters.

3. *If I understand correctly, the model estimates latent infection times, and there is also some data on symptom reporting for participants. Is it therefore possible to obtain an estimate of how many infections are asymptomatic? The values in Table S6 are difficult to generalise given that they depend on the study design, but it would be interesting if the model could provide an estimate of how many infections are likely to go undetected, and how serological dynamics relates to symptom reporting.*

Symptoms are only reported for the PCR-confirmed infections. Because we lack consistent symptom surveillance over the study period, we cannot relate the dynamics of infection and serology to symptom reporting. For our purposes, the importance of the symptom reporting was to rule out any clear difference in the magnitude of the titer boost between symptomatic and asymptomatic PCR-confirmed infections. We found none.

4. *In Fig S3, there is quite a lot of variation in the likelihood as the parameters vary. In some cases for H3N2, there is a difference of almost 10 log-likelihood points for similar parameter values. Is this an issue with convergence of the MIF algorithm? Or something to do with the structure of the data/model? Such differences are potentially concerning, given that some of the models compared in Table 1 only have a 20-25 difference in AICc.*

The variation in likelihood among profile points results from high Monte Carlo error, an implicit feature of stochastic inference. Please see our reply to Referee 1, point 11. We have no evidence of convergence problems with the MIF algorithm, as our requirements for convergence of multiple MIF searches from the same starting parameters were consistently met and the MCAP technique returned smooth profiles adjusted for the MC error.

Referee 2 remarked upon the difficulty of interpreting the results of model selection in the face of high Monte Carlo error. A benefit of the new weighted susceptibility model is that the parameter estimate for the weight of HI-correlated protection, $\psi_{a_i,s}$, reveals the best-fit form of protection (HI-correlated, non-HI-correlated, or a combination). Therefore, there is only one model, and we no longer use AICc to compare different forms of protection.

5. *How well did the model fit the individual-level titres? Figs S7 - S8 compare the distributions of rises, but it would be good to see a histogram of observed minus estimated titres at time of sampling across all participants.*

Please see our reply to Referee 1, point 3.

6. The combined susceptibility function in Equation 3 seemed a bit inelegant, given that it can produce an abrupt switch in the dominant component of protection (as seen in Fig 2). Did the authors consider a function with a smoother transition, like a weighted average of the two?

We have implemented Referee 2's suggestion to use the weighted average of HI-correlated and non-HI-correlated protection (Eq. 4). As described in "Summary of new results", this more concise representation of susceptibility, with nested models of purely HI-correlated and non-HI-correlated protection, yields the same major results as the comparison of discrete models. We believe that this representation of protection is better. It implicitly incorporates the three proposed forms of protection (HI-correlated vs. non-HI-correlated vs. both) in the value of $\psi_{a_i,s}$, the weight of HI-correlated protection for an individual of age a_i (children or adults) against subtype s , while providing a smooth trajectory for the post-infection dynamics of susceptibility $q_{i,s}(t)$.

7. What was the logic for normalizing titre boosts by the gap between samples in Fig S15?

We revised section S1 of the Supporting Information to clarify:

We compared the distributions of titer changes between symptomatic and asymptomatic infections and between primary and secondary infections (Fig. S14). Because titers wane, we normalized the boost for the interval between the pre- and post-infection sample dates.

Minor comment

1. It took me a while to work out precisely what the authors meant by infection, susceptibility and protection in the model. As I understand it, they define infection as an exposure to influenza that generates a detectable antibody response, susceptibility is the probability an exposure will result in such an infection, and nonsusceptibility is equivalent to protection. However, it would be helpful to explicitly define these terms (e.g. protection could be interpreted as protection from disease rather than serological response).

We have clarified the definitions of susceptibility in our revised "Complete Model Description" (lines 237 - 315):

(lines 257 - 260) an individual's instantaneous infection rate or force of infection, $\lambda_{i,s}(t)$, is determined by the rate of exposure to infectious contacts in the community and household and by the individual's susceptibility, defined as the probability of infection per infectious contact. Infections occur stochastically and can change the latent titers.... (lines 289 - 291) Individual i 's susceptibility to subtype s at time t , $q_{i,s}(t)$, is defined as the probability of infection given contact with an infected. Complete protection corresponds to $q_{i,s}(t) = 0$, complete susceptibility to $q_{i,s}(t) = 1$, and $0 < q_{i,s}(t) < 1$ to partial protection.

1 Referee 3

Remarks to the author

This manuscript describes the dynamics of protective immunity to influenza virus infection in a household cohort study. The main finding is that protective immunity in children is strongly associated with antibody titer (HI) levels, whereas the protective immunity in adults is weakly associated with antibody titer levels. The data set is unique and allows for a much deeper understanding of the immunity dynamics than before. The statistical modelling of the data is complex, which is inherent to the complexity of the system being modelled here. Overall, this manuscript presents an impressive effort towards a better understanding of protective immunity against influenza virus infection. However, there is an important issue that needs to be addressed. It is not clear that the age-specific exposure to influenza virus is modelled correctly. An individual's hazard rate of exposure is modelled as the product of an age-specific susceptibility per infectious contact event, the age-specific rate of contact events, times the probability that the contact is made with someone who is infectious. (in the manuscript the term risk is used, but I believe this must be hazard rate). The probability that contact is made with someone who is infectious is proportional to the proportion of ILI consultations among all GP consultations. The age-specific contact rates are based on observed social contact data (reference 49). The model ignores the age of the infectors. This introduces an inconsistency: it is supposed that the incidence of influenza virus infection of infectees varies with age, whereas the incidence of influenza virus infection of infectors does not vary with age. What I would have expected is the use of the observed contact rate matrix and a vector that reflects the typical distribution of infections over age times a time-varying function that describes the intensity of influenza circulation. Without this, the model includes an inherent inconsistency in the age-specific exposure to influenza virus. The key argument of the manuscript requires the age-specific exposure to be correct. Please inspect the model, correct it where needed, and discuss alternative explanations of the findings.

We have revised the model to address Referee 3’s concern about the age-specific distribution of the community influenza intensity. Previously, the model assumed that the hazard rate of community exposure varied among individuals only by differences in the age-specific number of daily contacts. In line with Referee 3’s suggestion, individual i ’s rate of exposure is now determined by the time-varying community flu intensity, the contact matrix describing the number of contacts among individuals of each age group, and the age distribution of infections among infectious contacts (estimated from Caini *et al.* 2018 *BMC Inf. Dis.*). The main conclusions are unchanged by the change in model structure (“Summary of new results”).

1.1 Minor comments

1. It is not clear from the text how susceptibility should be interpreted. From the methods section I understand it is a variable on a scale from 0 (complete immunity) to 1 (complete susceptibility). But what does a susceptibility of 0.3 mean? An individual with a susceptibility of 0.3 could have a probability of 0.3 of being completely immune, or this individual could have a probability of 0.3 being infected when exposed during each exposure?

We thank Referee 3 for raising this point too. Please see reply to Referee 2, Minor Comments, point 1.

2. The link is missing between the dynamics of the latent titer $h_{i,s}(t)$ and the likelihood function which is a function of $\log H_{i,s}$. Is the missing equation $\log H_{i,s} = \log_2(h_{i,s}/10) + 2$, or $\log H_{i,s} = \log(h_{i,s})$?

We have clarified in the main text (“Complete model description”):

(lines 326 - 332) The observed titer $H_{\text{obs},i,s}(t)$ relates to the corresponding latent titer $h_{i,s}(t)$ via the measurement model, which accounts for error in the titer measurements and the effect of discretization of titer data into fold-dilutions. The observed titers are fold-dilutions in the range [$<1:10$, $1:10$, $1:20$... , $1:5120$]. Consistent with other models (Zhao *et al.* 2017 *Am. J. Epidemiol.*, Yuan *et al.* 2017 *Epidemics*, Kucharski *et al.* 2015 *PLOS Biol.*), we define a log titer ($\log H$) for any titer h ,

$$\log H = \log_2\left(\frac{h}{10}\right) + 2, \quad (7)$$

such that the log of the observed titer takes on discrete values in the range [1,11]. In order to relate the observed titer $H_{\text{obs},i,s}(t)$ to the latent titer, $h_{i,s}(t)$, we begin by transforming both into log titers (Eq. 7), yielding the log observed titer $\log H_{\text{obs},i,s}(t)$ and the log latent titer $\log H_{i,s}(t)$. We assume that the observed log titer against subtype s is normally distributed around the log latent titer,

$$\log H_{\text{obs},i,s}(t) \sim \mathcal{N}(\log H_{i,s}(t), \epsilon), \quad (8)$$

where ϵ gives the standard deviation of the measurement error.

3. The text could be made more precise. For example: “epidemic incidences range from 5-17% in adults” requires a statement about the time unit for measuring incidence. The text and the equations do not always match. For example as “an exponential function that starts at 0” and “the individual trajectories in fig 2 (A,B) are constrained from above by the exponential curve” but equation 8 that the function is $1-\exp()$ rather than $\exp()$.

We thank Referee 3 for highlighting opportunities to improve the wording.

- (a) Previously “epidemic incidences range from 5-17% in adults”:

Revised text (lines 118 - 122): The incidences of individual H1N1pdm09 epidemics range from 7-12% in adults and 10-17% in children (Table 2). For H3N2, the incidences range from 4-12% in adults and 6-23% in children. Estimates of seasonal influenza incidence in the United States are 5-20% based on combined serology and viral infection (of influenza A and B) (Sullivan *et al.* 1993 *Am. J. Pub. Health*) and 3-11% based on symptomatic PCR-confirmed infections of influenza A (Tokars *et al.* 2018 *Clin. Inf. Dis.*). Converted to annual rates, the simulated incidences are similar, ranging from 5-17% in adults and 7-29% in children.

- (b) Previously “an exponential function that starts at 0”:

Revised text (line 52) Susceptibility determined by non-HI-correlated protection, $q_{2_{i,s}}(t)$, is a function that starts at 0 (no susceptibility) immediately after infection. The susceptibility increases as protection wanes exponentially at rate $w_{\text{nonspecific},i,s}$... (line 54) Titers in this model are still informative as indicators of infections, but they do not affect infection risk.

(c) Previously “the individual trajectories in fig 2 (A,B) are constrained from above by the exponential curve”

Revised text (lines 94 - 95) When we plot the latent susceptibility after each infection with each subtype, we see that the individual trajectories in Fig. 1A and B follow the curve describing the change in non-HI-correlated protection (Eq. 10).

We have revised throughout to increase clarity and precision.

REVIEWERS' COMMENTS:

Reviewer #1 (Remarks to the Author):

I have read the revised manuscript by Ranjeva and colleagues and the accompanying rebuttal letter. I have not, in this round of revisions, checked all formulas and methodology of the new extended models due to time limitations.

Overall, I am quite happy with the revised draft of the manuscript. In particular, I agree with the authors (p2 of the rebuttal letter) that this work is important because (1) it shows that cross-subtype protection is weak or even absent, (2) it is probably one of very few studies (perhaps the only?) that focuses on medically unattended infections which make up the bulk of all infections, (3) shows clearly that there are differences between children and adults in how well the HI titer correlates with protection to infection. I also found the text in the first paragraphs on p2 of the rebuttal letter on the roles of the head vs stalk of HA illuminating. The exposition is, I believe, actually better in the rebuttal letter than in the first paragraph of the discussion which I find a bit cryptic. I would expand the first paragraph of the discussion, paying special attention to explaining that (1) HI actually mainly measures Abs to the dominant 4/5 epitopes of the head of HA (and hardly takes into account stalk-directed Abs), and (2) where it is found that time since last infection is the better predictor for susceptibility in adults, it may actually be that this results from a mismatch of the head antibodies to evolved circulating viruses. This would also explain why half-life of protection against the faster evolving H3N2 is shorter than against the slower evolving H1N1.

This work is quite complex, and it is not easy, even for insiders, to fully grasp details of the analyses. Therefore, I believe it is important to be as transparent as possible, providing all data and code for future checking and additional analyses. I commend the authors for making everything available on github. I have had a quick look at the repository and believe that the repository may profit from some further editing, e.g., by providing a step-by-step explanation in the readme how the figures and tables can be reproduced.

Reviewer #2 (Remarks to the Author):

The authors have satisfactorily addressed my previous comments. In particular, the addition of a weighted function for HI-correlated protection has really strengthened the results.

We are grateful to the referees for their careful attention to this manuscript. We have addressed the remaining comments of Reviewer 1, which we detail in a point-by-point response below (responses in blue). Similarly, the revised text appears in blue in the manuscript pdf.

REVIEWERS' COMMENTS:

Reviewer #1 (Remarks to the Author):

I have read the revised manuscript by Ranjeva and colleagues and the accompanying rebuttal letter. I have not, in this round of revisions, checked all formulas and methodology of the new extended models due to time limitations.

Overall, I am quite happy with the revised draft of the manuscript. In particular, I agree with the authors (p2 of the rebuttal letter) that this work is important because (1) it shows that cross-subtype protection is weak or even absent, (2) it is probably one of very few studies (perhaps the only?) that focuses on medically unattended infections which make up the bulk of all infections, (3) shows clearly that there are differences between children and adults in how well the HI titer correlates with protection to infection. I also found the text in the first paragraphs on p2 of the rebuttal letter on the roles of the head vs stalk of HA illuminating. The exposition is, I believe, actually better in the rebuttal letter than in the first paragraph of the discussion which I find a bit cryptic.

I would expand the first paragraph of the discussion, paying special attention to explaining that (1) HI actually mainly measures Abs to the dominant 4/5 epitopes of the head of HA (and hardly takes into account stalk-directed Abs), and (2) where it is found that time since last infection is the better predictor for susceptibility in adults, it may actually be that this results from a mismatch of the head antibodies to evolved circulating viruses. This would also explain why half-life of protection against the faster evolving H3N2 is shorter than against the slower evolving H1N1.

We have accordingly expanded the introductory paragraphs of the Discussion (new text appears in bold below), which now read:

Our results suggest that protection against influenza A has different origins in adults and children. In children, the HI titer is a good correlate of protection, and infection durably boosts titers against H1N1pdm09. **In adults, time since infection is more strongly correlated with protection than the HI titer**, and infection is associated with small to no long-term changes in titer. **HI assays primarily measure antibodies to epitopes on the top of HA, and not epitopes on the sides of HA or on the stalk.** Thus, these results suggest that children tend to produce antibodies that target the head of the HA, which could mediate protection, whereas adults rely on antibody responses to other sites or potentially other forms of immunity for protection. This model is consistent with the concepts of antigenic seniority and original antigenic sin. Antigenic seniority

refers to the phenomenon that individuals' highest antibody titers to influenza are to strains that circulated in childhood (Lessler 2012 *PLoS Biol*), and original antigenic sin is the process by which antibody responses to familiar sites are preferentially reactivated on exposure to new strains (Cobey 2017 *Curr Op Virol*, Francis 1960 *Proc Am Phil Soc*, De St Groth 1966 *J Exp Med*). With time, these familiar sites may be the ones that are most conserved. On HA, these sites would tend to be away from the fast-evolving epitopes near the receptor binding domain, and would not be readily detected by HI assays. Consistent with this view, epidemiological studies have shown that levels of stalk-directed antibodies increase with age (Park 2018 *MBio*, Miller 2013 *Sci Trans Med*), **and that vaccination with a partially antigenically novel influenza virus boosts responses to conserved sites, including the stalk (Krammer 2014 *CVI*, Nachbagauer 2014 *J Virol*, Andrews 2015 *Sci Trans Med*, Fonville 2014 *Science*)**. Our model shows that these differences are persistent and relate to protection. **In adults, the time since last infection better correlates with protection than HI titer because their antibodies to the top of the HA head tend to be antigenically mismatched. Compared to children, more of their protection derives from other responses.**

We estimated that protection in both children and adults wanes with an average half-life of 3.5-7 years, lasts longer against H1N1pdm09 than H3N2, and lasts slightly longer in adults compared to children against H3N2. These timescales are consistent with the estimated decay of immunity over 2-10 years due to antigenic evolution in population-level models (Bock-Axelsen 2014 *PNAS*, Du 2017 *Sci Trans Med*). **Responses may be more durable to H1N1pdm09 compared to H3N2, and in adults compared to children for H3N2, because the epitopes that are targeted are relatively more conserved.** In contrast to adults, the dependence of protection on HI titer in children leads to substantial variation in susceptibility over time (Fig. 2) This heterogeneity may well extend to adults but is difficult to identify without much longer time series or other immune assays. The models' tendency to overestimate individuals' titer variation over time suggests that important differences between individual responses could be missing (Supplementary Figs. 8 and 9). Longer observation periods and more complete observations of the immune response can help separate these factors from behavioral or environmental differences in infection risk.

This work is quite complex, and it is not easy, even for insiders, to fully grasp details of the analyses. Therefore, I believe it is important to be as transparent as possible, providing all data and code for future checking and additional analyses. I commend the authors for making everything available on github. I have had a quick look at the repository and believe that the repository may profit from some further editing, e.g., by providing a step-by-step explanation in the readme how the figures and tables can be reproduced.

In line with the reviewer's suggestion, we have expanded the documentation of the GitHub repository. The new "Workflow for Subtype-level Analysis" section of the ReadMe file provides a step-by-step overview of the model implementation, inference,

and analysis of the results. Each step of the overview provides a link to the appropriate subdirectory containing the code and instructions to generate the associated results and figures. The main ReadMe file also provides an in-depth guide to the maximum likelihood-based parameter estimation (the section titled “Running the Models to Estimate Parameters”).

Reviewer #2 (Remarks to the Author):

The authors have satisfactorily addressed my previous comments. In particular, the addition of a weighted function for HI-correlated protection has really strengthened the results.